# Federated Split Vision Transformer for COVID-19 CXR Diagnosis using Task-Agnostic Training

**Sangjoon Park**[1*]          **Gwanghyun Kim**[1*]

**Jeongsol Kim**[1]          **Boah Kim**[1]          **Jong Chul Ye**[1,2,3]

[1] Department of Bio and Brain Engineering
[2]Kim Jaechul Graduate School of AI, [3]Deptartment of Mathematical Sciences
Korea Advanced Institute of Science and Technology (KAIST)
{depecher, gwang.kim, wjdthf3927, boahkim, jong.ye}@kaist.ac.kr

## Abstract

Federated learning, which shares the weights of the neural network across clients, is gaining attention in the healthcare sector as it enables training on a large corpus of decentralized data while maintaining data privacy. For example, this enables neural network training for COVID-19 diagnosis on chest X-ray (CXR) images without collecting patient CXR data across multiple hospitals. Unfortunately, the exchange of the weights quickly consumes the network bandwidth if highly expressive network architecture is employed. So-called split learning partially solves this problem by dividing a neural network into a client and a server part, so that the client part of the network takes up less extensive computation resources and bandwidth. However, it is not clear how to find the optimal split without sacrificing the overall network performance. To amalgamate these methods and thereby maximize their distinct strengths, here we show that the Vision Transformer, a recently developed deep learning architecture with straightforward decomposable configuration, is ideally suitable for split learning without sacrificing performance. Even under the non-independent and identically distributed data distribution which emulates a real collaboration between hospitals using CXR datasets from multiple sources, the proposed framework was able to attain performance comparable to data-centralized training. In addition, the proposed framework along with heterogeneous multi-task clients also improves individual task performances including the diagnosis of COVID-19, eliminating the need for sharing large weights with innumerable parameters. Our results affirm the suitability of Transformer for collaborative learning in medical imaging and pave the way forward for future real-world implementations.

## 1 Introduction

After its earlier success in many fields, deep neural networks have found a pervasive suite of applications in healthcare research including medical imaging, becoming a new de facto standard [54, 19, 13, 49, 17, 63, 6, 58, 24]. Training these networks requires a vast amount of data to achieve robust performance [8, 11, 47]. Despite the fact that multi-center collaboration is mandatory due to the shortage of labeled data in a single institution, collaboration in healthcare research is heavily

---

[*]Authors contributed equally.

35th Conference on Neural Information Processing Systems (NeurIPS 2021).

impeded by difficulties in data sharing stemming from the privacy issues and limited consent of patients [50, 38, 53].

To alleviate this problem, the distributed machine learning methods, devised to enable the computation on multiple clients and servers leaving data to reside on the source devices, can be effectively leveraged for healthcare research [7, 43]. Federated learning (FL) is one of these methods which enables model training on a large corpus of decentralized data [26, 32, 59]. However, FL still holds several limitations in that it depends on clients' computational resources for its client-side parallel computation strategy for update and is not free from privacy concerns [27, 31, 52]. In contrast to FL, another distributed machine learning method, split learning (SL) offers better privacy and requires lower computational resources of clients by splitting the network between clients and the server [22, 52], but still possess problems that it shows significant slower convergence than FL and can not learn under non-independent and non-identically distributed (non-IID) data [20].

Especially under unprecedented pandemic of an emerging pathogen like COVID-19, under which direct multi-national collaboration is deterred for prevention of epidemics, the collaboration via these distributed machine learning approaches is becoming increasingly important, since these enable to build a model with performance tantamount to data-centralized learning without any direct sharing of raw data between institutions to offer privacy.

Recently proposed Vision Transformer (ViT) architecture [14], inspired by astounding results of Transformer-based models on natural language processing (NLP), have demonstrated impeccable performance on many vision tasks by enabling to model long dependencies within images. Besides this strength, the straightforward design of the Transformer allows to easily decompose the entire network into parts: the head for extracting features from the input image, the Transformer body to model the dependency between features, and the tail used for mapping features to task-specific output. One of the important contributions in this paper is the observation that this configuration is optimal for SL where a network should be split into the parts for clients and servers. In addition, as suggested in [9], the Transformer body with sufficient capacity can be shared between various tasks, being suitable for multi-task learning (MTL) to leverage robust representation from multiple related tasks to enhance the generalization performance of individual tasks.

Accordingly, here we propose a novel Federated Split Task-Agnostic (FeSTA) framework equipped with a Transformer to simultaneously process multiple chest X-ray (CXR) tasks including diagnosis of COVID-19, emulating a real collaboration between several hospitals. To validate the practicability of FeSTA, we also implemented the framework using a friendly federated learning framework (Flower) protocol [2], confirming seamless integration of various components. Experimental results show that our framework can show stable performance even under non-IID settings which is a frequently faced situation for collaboration between hospitals while offering privacy, by amalgamating FL and SL to maximally exploit their main advantages. In addition, we show that the proposed FeSTA Transformer along with MTL improves the performances of individual tasks. In summary, our contributions are two folds:

- We proposed a novel FeSTA learning framework equipped with the ViT by utilizing its decomposable design to amalgamate the merit of FL and SL.

- We showed that the model trained with the FeSTA framework can leverage the robust representations from multiple related tasks to improve the performance of the individual task.

## 2   Related Work

**Distributed machine learning.**   FL is a distributed machine learning approach originally proposed by Google [26] to enable the training of a model via distributed devices and data. It eliminates the need to aggregate the raw data in a centralized way by enabling the model to be updated on the edge devices (e.g. mobile phones, computers in hospitals). Specifically, during the training, the server initializes a global model and sends it to each client. The clients then train the model with their local data in parallel and return the updated model to the server. Then, the server aggregates and distributes those models by a method such as federated averaging (`FedAvg`) [32] to update the global model. This process (called round) continues repeatedly until the model converges. Though FL enables

decentralized training in a privacy-preserving manner, it still holds limitations as it largely depends on the computation resources of clients and is vulnerable to model inversion attacks [56].

Different from FL, SL divides the neural network into several sub-networks, and these separated sub-networks are trained under distributed setting [16]. In detail, the first sub-network is trained on the client-side with local data and then passes the feature to the second sub-network located in the server. The server can access the only feature from the first sub-network, and train the second sub-network to send the subsequent feature to the third sub-network on the client. Finally, the third sub-network is able to provide the output of the overall network. By inserting black-box sub-networks in both client and server sides, it is possible to offer better privacy than FL. Besides the privacy benefit, SL uses less computational resources than FL at the client-side. Nevertheless, since the one cycle of forward and backward passes is finished after data and gradients move back and forth across the sub-networks distributed on multiple sides, the convergence of SL is considerably slower than FL. In addition, it was reported that the convergence is not reached at all under the non-IID setting [20].

**Multi-task learning.** MTL is a learning strategy to improve the generalization performance of a specific task with the help of information from other related tasks. In MTL, models for multiple related tasks are trained simultaneously [5, 16]. In its early era, the motivation of MTL is to mitigate the data insufficiency problem where the number of labeled data is limited for each task to train an accurate learner. MTL helps to reuse knowledge and thereby reduce the requirement for labeled data for each task, exhibiting that the MTL model can achieve better performance than the single-task counterpart in many fields ranging from computer vision [15, 34, 36, 28] to NLP [30, 21, 29, 42]. With increased data, the MTL model can learn more robust representations via knowledge sharing among multiple tasks, resulting in improved performance and less overfitting for the individual task.

**Vision Transformer.** Transformer [51], which was originally developed for NLP, is a deep neural network based on an attention mechanism that utilizes an appreciably large receptive field. After achieving state-of-the-art (SOTA) performance in NLP, it has also inspired the vision community to explore its application on vision tasks to utilize its ability to model long-range dependency within an image [25]. The ViT was one of the successful attempts to apply Transformer directly to images, achieving excellent results compared to the SOTA convolutional neural networks in image classification tasks [14]. Furthermore, in addition to the superb performance, the straightforward modular design of ViT facilitates broad applications in many tasks only with minimal change. Chen et al. [9] proposed image processing transformer, which is one of the successful multi-task model for various computer vision tasks, by splitting ViT into shared body and task-specific heads and tails, suggesting that Transformer body with sufficient capacity can be shared across relevant tasks. However, they leveraged encoder-decoder design and the usefulness of the multi-task ViT model was not evaluated along with the distributed learning methods.

Recently, ViT was successfully used for diagnosis and severity prediction of COVID-19, showing the SOTA performance [35]. Specifically, to alleviate the overfitting problem with limited data available, the overall framework is decomposed into two steps: the pre-trained backbone network to classify common low-level CXR features, which was leveraged in the second step by Transformer for high-level diagnosis and severity prediction of COVID-19. By maximally utilizing the merit of the large-scale database containing more than 220,000 CXR images, the model has attained stable generalization performance as well as SOTA performance in a variety of external test data from different institutions, even with the limited number of labeled cases for COVID-19.

## 3 Split Task-Agnostic Transformer for CXR COVID-19 Diagnosis

Inspired by these works, here we are interested in utilizing ViT for distributed learning in COVID-19 CXR diagnosis, where the collaboration via these distributed machine learning approaches is becoming increasingly important by offering privacy and still allowing similar performance to the data-centralized learning.

The reason we are interested in ViT architecture for this purpose is that the natural configuration of ViT may be optimal for MTL as well as SL where the easily decomposable modular design of the network is preferred, suggesting a possibility to maximally reconcile the merits of MTL and SL through ViT architecture. Specifically, the clients just train the head and tail parts of the network, whereas the Transformer body is shared across multiple clients. Then, the embedded features from

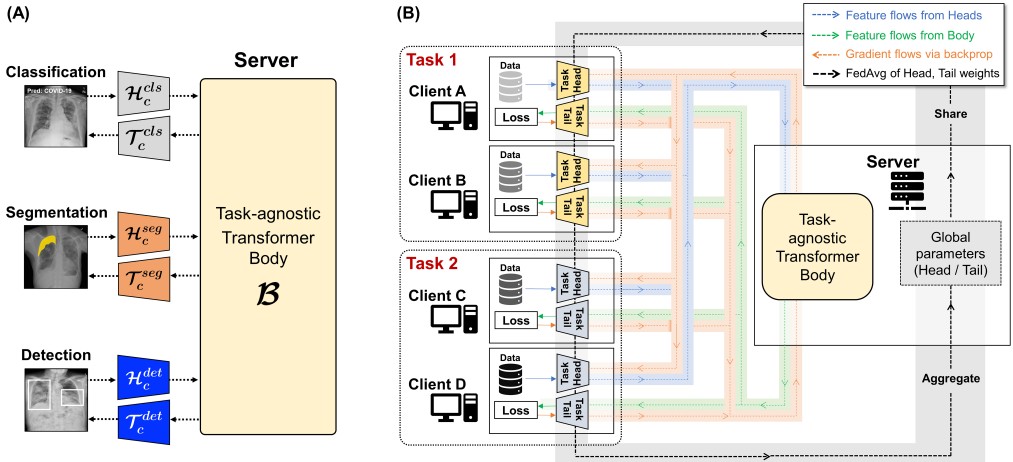

Figure 1: Overall framework of the proposed method. (a) Split task-agnostic CXR Transformer for multi-task learning and (b) the concept of FESTA learning process.

the head network from multiple clients can be leveraged in the second step by Transformer to process individual tasks including diagnosis of COVID-19. Furthermore, by maximally utilizing the merit of the large-scale database from CXR for various tasks, our goal is to demonstrate stable generalization performance as well as SOTA performance in the external test dataset.

### 3.1 FESTA: Federated Split Task-Agnostic Learning

The concept of the proposed Federated Split Task-Agnostic (FESTA) learning process is illustrated in Fig 1(a) and (b). Let $\mathcal{C} = \bigcup_{k=1}^{K} C_k$ be a group of client sets with different CXR tasks, where $K$ denotes the number of tasks and $C_k$ has one or more clients with different datasets each other for the $k$-th task, i.e. $C_k = \{c_1^k, c_2^k, \ldots, c_{N_k}^k : N_k \geq 1\}$. Each client $c \in C_k$ has its own task-specific network architecture for a head $\mathcal{H}_c$ and a tail $\mathcal{T}_c$, which are connected to Transformer $\mathcal{B}$ in the server.

In our FeSTA framework, the server first initializes the weights of the Transformer body and task-specific heads, tails for each task $k$. Then, it distributes the initialized weights of heads and tails to each client $c \in C_k$. For round $i = 1, 2, \ldots R$, each client (e.g. hospital) perform the forward propagation on their task-specific head and pass their intermediate feature to the server. Specifically, using the local training data $\{(x_c^{(i)}, y_c^{(i)})\}_{i=1}^{N_c}$, the head network $\mathcal{H}_c$ encodes the smashed feature maps $h_c^{(i)}$ and sent it to the server: $h_c^{(i)} = \mathcal{H}_c(x_c^{(i)})$. Then, the server-side Transformer body $\mathcal{B}$ receives the feature from the all clients and generate the features $b_c^{(i)}$ for $c$ in parallel: $b_c^{(i)} = \mathcal{B}(h_c^{(i)})$. Resulting smashed features from the Transformer are allocated to task-specific tail to produce the final prediction $\hat{y}_c^{(i)} = \mathcal{T}_c(b_c^{(i)})$ according to each task, and the forward path finishes. Subsequently, the loss for each client can be calculated as $\ell_c(y_c^{(i)}, \mathcal{T}_c(\mathcal{B}(\mathcal{H}_c(x_c^{(i)}))))$ where $\ell_c(y, \hat{y})$ refers to the $c$ client- specific loss between the target $y$ and the estimate $\hat{y}$. By minimizing the loss with respect to the tail weight, the gradients of the local tails are passed reversely to the server. After receiving the gradients, the server performs the back-propagation on the server-side body model, sends the gradients back to the clients.

Specifically, for the task-agnostic body update, the following optimization problem is solved:

$$\min_{\mathcal{B}} \sum_{c \in \mathcal{C}} \sum_{i=1}^{N_c} \ell_c(y_c^{(i)}, \mathcal{T}_c(\mathcal{B}(\mathcal{H}_c(x_c^{(i)})))), \tag{1}$$

For the task-specific fine-tuning, the following optimization problem is solved:

$$\min_{\mathcal{H}_c, \mathcal{T}_c} \sum_{i=1}^{N_c} \ell_c(y_c^{(i)}, \mathcal{T}_c(\mathcal{B}(\mathcal{H}_c(x_c^{(i)})))), \tag{2}$$

Finally, the server aggregates and averages the weights of local heads and tails from the clients to update the global heads and tails among the clients for the same tasks via `FedAvg`, and distributes back the updated global weights of heads and tails for each task $k$ to the clients. The algorithm is formally presented in Algorithm 1.

---

**Algorithm 1:** FESTA: Federated Split Task-Agnostic learning

---

1  **Function** `ServerMain`:
2     Initialize the body weight $w_{\mathcal{B}}^{(1)}$ and client head/tail weights $(\bar{w}_{\mathcal{H},k}, \bar{w}_{\mathcal{T},k})$ for each task $k \in \{1, ..., K\}$ in server
3     **for rounds** $i = 1, 2, \ldots R$ **do**
4         **for tasks** $k \in \{1, 2, \ldots K\}$ **do in parallel**
5             **for clients** $c \in C_k$ **do in parallel**
6                 **if** $i = 1$ **or** $(i - 1) \in$ UnifyingRounds **then**
7                     Set client $(w_{\mathcal{H}_c}^{(i)}, w_{\mathcal{T}_c}^{(i)}) \leftarrow (\bar{w}_{\mathcal{H},k}, \bar{w}_{\mathcal{T},k})$
8                 $h_c^{(i)} \leftarrow$ `ClientHead`$(c)$
9                 $b_c^{(i)} \leftarrow \mathcal{B}(h_c^{(i)})$
10                 $\frac{\partial L_c^{(i)}}{\partial b_c^{(i)}} \leftarrow$ `ClientTail`$(c, b_c^{(i)})$ & Backprop.
11                 $(w_{\mathcal{H}_c}^{(i+1)}, w_{\mathcal{T}_c}^{(i+1)}) \leftarrow$ `ClientUpdate`$(c, \frac{\partial L_c^{(i)}}{\partial h_c^{(i)}})$
12         Update body $w_{\mathcal{B}}^{(i+1)} \leftarrow w_{\mathcal{B}}^{(i)} - \frac{\eta}{K} \sum_{k=1}^{K} \sum_{c \in C_k} \frac{\partial L_c^{(i)}}{N_k \partial w_{\mathcal{B}}^{(i)}}$
13         **if** $i \in$ UnifyingRounds **then**
14             **for tasks** $k \in \{1, 2, \ldots K\}$ **do**
15                 Update $(\bar{w}_{\mathcal{H},k}, \bar{w}_{\mathcal{T},k}) \leftarrow (\frac{1}{N_k} \sum_{c \in C_k} w_{\mathcal{H}_c}^{(i+1)}, \ \frac{1}{N_k} \sum_{c \in C_k} w_{\mathcal{T}_c}^{(i+1)})$

16  **Function** `ClientHead`$(c)$:
17     $x_c \leftarrow$ Current batch of input from client $c$
18     **return** $\mathcal{H}_c(x_c)$
19  **Function** `ClientTail`$(c, b_c)$:
20     $y_c \leftarrow$ Current batch of label from client $c$
21     $L_c \leftarrow \ell_c(y_c, \mathcal{T}_c(b_c))$ & Backprop.
22     **return** $\frac{\partial L_c}{\partial b_c}$
23  **Function** `ClientUpdate`$(c, \frac{\partial L_c}{\partial h_c})$:
24     Backprop. & $(w_{\mathcal{H}_c}, w_{\mathcal{T}_c}) \leftarrow (w_{\mathcal{H}_c} - \eta \frac{\partial L_c}{\partial w_{\mathcal{H}_c}}, \ w_{\mathcal{T}_c} - \eta \frac{\partial L_c}{\partial w_{\mathcal{T}_c}})$
25     **return** $(w_{\mathcal{H}_c}, w_{\mathcal{T}_c})$

---

### 3.2 Multi-task CXR learning

For synergistic performance improvement, we explore the following three tasks that are commonly used for CXR: classification, segmentation, and object detection. These three tasks were separately trained for the individual model, while used simultaneously to train and to evaluate the task-agnostic model for multiple related tasks. The details of each task and dataset are as follows.

**COVID-19 classification.** This is the main task we want to achieve through FESTA. Since we initialized the weights of classification heads to be the robust feature extractor trained on pre-built large data corpus containing common CXR findings, the classification heads were initialized with the pre-trained weights from CheXpert [23] dataset containing 10 CXR findings (no finding, cardiomegaly, opacity, edema, consolidation, pneumonia, atelectasis, pneumothorax, pleural effusion, support device) labeled by experts. From pre-training on the CheXpert dataset, we excluded 32,387 lateral view images, and 29,420 posterior-anterior (PA) and 161,427 anterior-posterior (AP) view data were finally used. Table 1 summarizes dataset resources and partitioning for the classification task. We used both public datasets containing labels of infectious disease (Valencian Region Medical

Table 1: Datasets and sources for COVID-19 diagnosis

| Total CXR images | External | Training and validation dataset | | | | | |
|---|---|---|---|---|---|---|---|
| | | Client 1 | Client 2 | Client 3 | Client 4 | Client 5 | Client 6 |
| | Hospital 1 | Hospital 2 | Hospital 3 | Hospital 4 | NIH | Brixia | BIMCV |
| Normal | 13,649 | 320 | 300 | 400 | 8,861 | 3,768 | - | - |
| Other infection | 1,468 | 39 | 144 | 308 | 977 | - | - | - |
| COVID-19 | 2,431 | 6 | 8 | 80 | - | - | 1,929 | 408 |
| **Total CXR** | **17,548** | **365** | **452** | **788** | **9,838** | **3,768** | **1,929** | **408** |

Image Bank [BIMCV] [12], Brixia [44, 4], National Institutes of Health [NIH] [55]), and CXR data deliberately collected from four hospitals labeled by board-certified radiologists. We put one hospital data aside as an external test dataset to evaluate the classification performance for real-world applications. Overall, 17,183 PA view CXR images were used for training/validation and 365 PA view CXR images for the test. In addition, we performed the experiments in the view-agnostic setting by adding AP view CXRs, to further extend real-world applicability as provided in Appendix C.1. By adding AP view data, the total amounts of CXRs were increased from 17,183 to 24,180 for training/validation and 365 to 556 for external test datasets. Notably, the number of COVID-19 cases was increased from six to 81 in the external test dataset. We modeled data from different sources as individual clients, emulating the real-world collaboration between hospitals. This setting is important to validate our method in non-IID data distribution. The study was ethically approved by the Institutional Review Board at each participating hospital and the requirement for informed consent was waived.

**Segmentation.** For the segmentation task, we have used the Society for Imaging Informatics in Medicine and the American College of Radiology Pneumothorax Segmentation Challenge dataset [45] consisting of 12,047 CXR images for training, 3,205 images for testing. The training dataset was divided randomly with a 4:1 ratio into training and validation datasets, and we developed and fine-tuned the model for pneumothorax segmentation with these datasets. Afterward, the segmentation performance of the model was evaluated in testing datasets. Since the data for segmentation was anonymized, it was not possible to divide the dataset according to the sources of acquisition. Instead, we randomly divided the entire dataset into the two clients, emulating the collaboration between two hospitals.

**Object detection.** For the object detection task, the model was constructed to detect lung opacities in CXR with the Radiological Society of North America (RSNA) Pneumonia Detection Challenge dataset [41] that consists of CXR images and labels with bounding boxes and detailed class information of 26,684 subjects. We randomly divided the entire dataset with a 3:1 ratio into training and testing datasets, with which model was trained and evaluated. Similar to the segmentation task, as the data contains no information about the sources of acquisition, the dataset was randomly divided into two clients to simulate the collaboration between hospitals.

## 4 Experimental Results

### 4.1 Implementation details

For the head and tail parts of our model, the networks specialized for each task were utilized. For classification, we used the modified version of the network proposed by Ye et al. [60], which comprises DenseNet combined with Probabilistic Class Activation Map (PCAM) operations for the classification task, since it achieved outstanding performance in the CXR classification competition. For segmentation, we adopt TransUNet [10] tthat inserts a Transformer between the convolutional neural network (CNN) encoder and decoder of UNet [40] to take advantage of both architectures. Similarly, the 2nd place solution in the RSNA pneumonia detection challenge, which is a modified version of RetinaNet for object detection, was utilized. After splitting these models into head and tail, the feature maps between head and tail were mapped to the feature maps with the same dimension of $16 \times 16 \times 768$, and used as input of Transformer body. The Transformer body, equipped with 12 layers of standard Transformer encoder with 12 attention heads, transforms the feature maps to

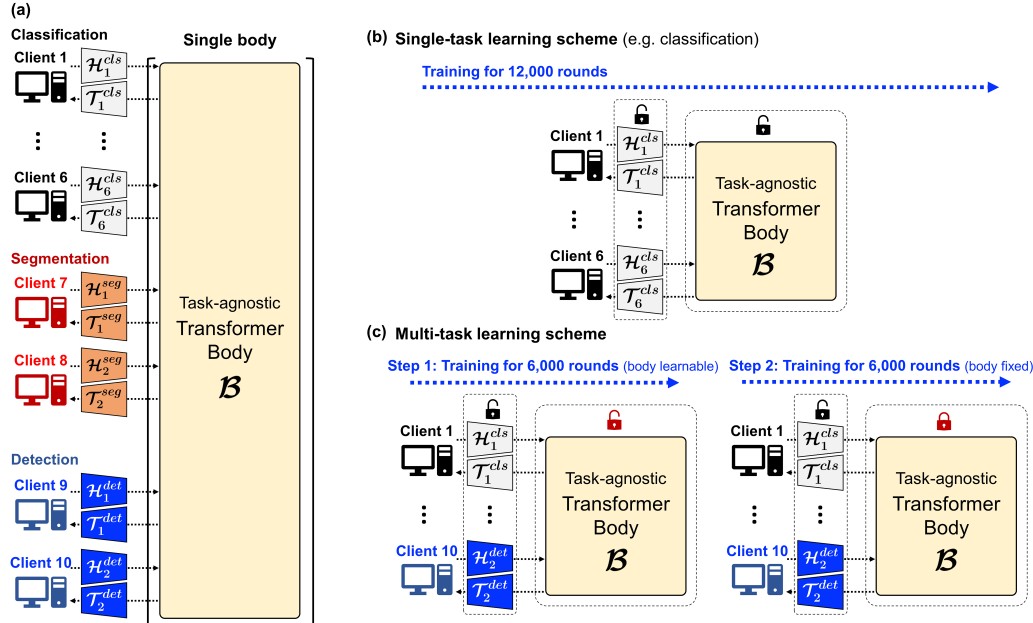

Figure 2: Implementation details of the proposed method. (a) Detailed experimental setting for multi-task learning (MTL) with three different tasks composed of 10 clients (six for classification, two for segmentation, two for detection. Training scheme for (b) single-task learning, and (c) MTL.

embed better representation. Then, the resulting feature maps from the body were passed to tails to yield the outcomes.

As suggested in Park et al. [35], the head for classification was first initialized with pre-trained weights from the CheXpert dataset. We minimized the cross-entropy loss for the classification task. For the segmentation model, we minimized the binary cross-entropy loss combined with dice and focal loss. Finally, for the detection task, we minimized the sum of box classification, box regression, and image classification losses as suggested by Gabruseva et al. [18]. For all tasks, the batch size was 2 per client, and the warm-up step was 500. We set the number of total rounds to 12,000, and the weights of each clients' head and tail underwent `FedAvg` per 100 rounds by the server.

Fig. 2 illustrates the implementation details and the experimental settings of the proposed method. To simulate single-task learning (STL) in Fig. 2(b), we considered each six data sources as different clients for the classification task. On the other hand, for MTL shown in Fig. 2(c), 10 clients were simultaneously used, with six clients for classification and two clients for segmentation and detection tasks, respectively. For the segmentation and detection tasks, the training data set was randomly split into two subsets, whereas non-IID distribution was used for the classification task. To adjust the loss scale, the customized weights of 1:2:2 were applied for classification, segmentation, and detection tasks to update the common body weights. We divided the MTL into two steps, jointly training the task-specific heads, tails, and the task-agnostic body (6,000 rounds), and fine-tuning only the task-specific heads, tails with the body weights fixed (6,000 rounds). By fixing the parameters of the shared Transformer body during the second step, the best models can be selected for different tasks according to the performance evaluation metrics of each task, even at the different rounds.

The FeSTA, FL, and SL simulation was performed on the modified version of Flower (licensed under an Apache-2.0 license) [2] FL framework. All experiments were performed with Python version 3.8 and Pytorch version 1.7 on Nvidia RTX 3090, 2080 Ti, and 1080 Ti. For more details of implementation, refer to Appendix A.

**Performance metrics.** We used the area under the receiver operating characteristic curve (AUC) to evaluate the diagnostic performance in the classification task. To evaluate the accuracy of segmentation, the Dice coefficient was used to quantitatively measure the overlap between the segmentation results by model and the ground truths. The detection results were evaluated by calculating the mean

average precision (mAP) at the different intersection over union, with a threshold range from 0.4 to 0.75 with a step size of 0.05 as suggested in RSNA [41]. All experiments have been run and evaluated with three different random seeds for the weight initialization to prevent the chance of confusing the results.

## 4.2 Results

**FESTA vs. Other strategies.** We compared the performance of classification model trained with FESTA with a data-centralized setting and other distributed learning strategies for COVID-19 classification task under the non-IID setting. To simulate data-centralized training, the whole network with connected head, body, and tail was trained with the integrated dataset of all six sources. To simulate FL, the whole network was aggregated and distributed by the server with `FedAvg` as suggested by McMahan et al. [32]. On the other hand, for SL, the split sub-networks reside in the clients and server-side, and the client-side sub-networks were not aggregated as in [52]. The same experimental settings and hyperparameters were applied for a fair comparison. For more details of the data-centralized learning and other distributed learning methods, refer to Appendix B.

As shown in Table 2, our method achieved comparable performance to the data-centralized learning method as well as outperformed the existing distributed learning methods, suggesting the superiority of our method over other methods. Of note, the performance could be further enhanced with MTL, which is a distinct strength of the proposed method.

In our additional experiments after adding AP view data, the model showed even better performance with the increased number of cases (Appendix C.1). This remarkable view-agnostic behavior of the model incentivizes the real-world application of computer-aided diagnosis of COVID-19, as the recent review on artificial intelligence models for COVID-19 diagnosis claims that the AI model, which has been pouring out a lot recently, are not helpful at all from the viewpoint of clinical application [57], in which the performances were substantially unstable by the factors like the view of CXR images [37].

Table 2: Comparison for the performance of the proposed method with other strategies

| Strategy | AUC | | | |
| --- | --- | --- | --- | --- |
| | Average | COVID-19 | Others | Normal |
| Data-centralized | $0.911 \pm 0.016$ | $0.883 \pm 0.036$ | $0.927 \pm 0.013$ | $0.923 \pm 0.004$ |
| Federated learning | $0.891 \pm 0.019$ | $0.840 \pm 0.035$ | $0.926 \pm 0.018$ | $0.906 \pm 0.028$ |
| Split learning | $0.863 \pm 0.005$ | $0.807 \pm 0.012$ | $0.892 \pm 0.007$ | $0.889 \pm 0.019$ |
| FESTA (single-task learning) | $0.909 \pm 0.021$ | $0.880 \pm 0.008$ | $0.916 \pm 0.038$ | $0.931 \pm 0.021$ |
| **FESTA (multi-task learning)** | **$0.931 \pm 0.004$** | **$0.926 \pm 0.023$** | **$0.929 \pm 0.016$** | **$0.938 \pm 0.013$** |

Note: Experiments were performed repeatedly with three random seeds to report mean and standard deviation. For evaluation of split learning, the average metric between clients are calculated.

**Multi-task learning vs. Single-task learning.** We evaluate whether the task-agnostic Transformer body contributes to improving the performance of the entire model by leveraging better representation from the MTL with several related tasks, namely classification, segmentation, and object detection in this work. As provided in Table 3, the models trained with the MTL approach showed at least comparable or even better performance compared with STL counterparts without the need to create an individual body model for each task or sharing a large body model between clients, which suggest the distinct merit of MTL approach with our framework.

In addition to the STL models, the MTL models trained with our framework outperformed task-specific expert models and provided comparable or even better performances to Kaggle's winning solution for the same tasks. Of note, when we substitute the shared layers of the Transformer body with CNN architecture with similar complexity, the performance gain was no longer maintained, suggesting the suitability of Transformer architecture in learning shared representation between the related tasks as provided in the additional experiments in Appendix C.2.

**Model sizes and communicative benefit.** Table 4 shows the numbers of parameters and the sizes of sub-networks. The task-agnostic body is the largest among sub-networks in terms of both the number of parameters and model size. This suggests that the largest part of the model does not need to be aggregated and distributed between client-client and client-server, which offers substantial

Table 3: Comparison of the performances between single-task and multi-task learning

| Tasks | Metrics | Single-task learning | Multi-task learning |
|---|---|---|---|
| **Classification** | AUC | $0.909 \pm 0.021$ | **$0.931 \pm 0.004$** |
| **Segmentation** | Dice | $0.798 \pm 0.016$ | **$0.821 \pm 0.003$** |
| **Detection** | mAP | $0.202 \pm 0.008$ | **$0.204 \pm 0.002$** |

Note: Experiments were performed repeatedly with three random seeds to report mean and standard deviation.

communicative benefit. In all tasks, the size of the body is more than half of the entire network. Therefore, not having to share this huge part means the total training time can be reduced considerably. In our experiments, the training time with the proposed method was four times shorter than the FL approach in which the entire network is shared and distributed. In addition, the proposed method offers another benefit of saving the computational resources of the server by processing multiple tasks with a single body model to enable the server to efficiently handle the requests for various tasks from many clients simultaneously.

Nonetheless, there remains a concern of communicative heavy load between the server and clients caused by the frequent transmission of features and gradients. Our further analysis of communication costs including features and gradients transmission are provided in the additional experiments, which suggests that the communication costs of the proposed FESTA framework was substantially lower than those of FL, although higher than SL. For detailed results of the computation for communication costs, refer to Appendix C.3.

Table 4: Parameter numbers and model sizes of the sub-networks

| Task | Head | | Body | | Tail | |
|---|---|---|---|---|---|---|
| | **Parameters** | **Size** | **Parameters** | **Size** | **Parameters** | **Size** |
| **Classification** | 13.313 M | 54.1 MB | | | 0.002 M | 11.7 KB |
| **Segmentation** | 15.041 M | 60.2 MB | 66.367 M | 265.5 MB | 7.387 M | 29.6 MB |
| **Detection** | 27.085 M | 108.8 MB | | | 19.773 M | 79.1 MB |

Note: Model sizes were estimated by parameter numbers and file sizes of saved weights.

## 4.3   Ablation study

**Role of Transformer body.**   As provided in Table 5, we first performed the ablation study to verify the contribution of the Transformer body on the server. The model without the server-side Transformer body, which is identical to the DenseNet-121 model equipped with PCAM operation [60], was built and trained with the same setting, and compared with the proposed model containing the Transformer body. The AUC values were higher with the Transformer body either for STL or MTL compared to those without the Transformer body with the statistical significance, implying the advantageous role of the server-side Transformer body as in Appendix C.4.

**Numbers of round for averaging.**   Determining the optimal number of rounds for averaging is important. If the aggregation, averaging, and distribution, namely FESTA is performed too frequently, the cost and required resources for communication increases, which can interfere or even preclude the learning process. On the contrary, if averaged too rarely, naive averaging of the learned model parameters of local learners can be devastating, resulting in nonsensical parameters in some layers [61]. Therefore, we performed the ablation to determine the optimal number of rounds to conduct FESTA. As shown in Table 5, the averaging per 100 rounds showed the comparable or better performance to less or more frequent counterparts.

More results of ablation studies are provided in Appendix D.

## 5   Conclusions

In this paper, we proposed a novel Federated Split Task-Agnostic (FESTA) framework suitable to leverage the formidable benefit of ViT to simultaneously process multiple CXR tasks including the

Table 5: Role of Transformer body and round number for FESTA

| Method | AUC | | | |
| --- | --- | --- | --- | --- |
| | **Average** | **COVID-19** | **Others** | **Normal** |
| Role of Transformer body | | | | |
| w/o Transformer body | 0.889 ±0.015 | 0.874 ±0.056 | 0.895 ±0.011 | 0.898 ±0.009* |
| **w Transformer body** | **0.909 ± 0.021** | **0.880 ± 0.008** | **0.916 ± 0.038** | **0.931 ± 0.021*** |
| Number of rounds for FESTA | | | | |
| per 10 rounds | 0.822 ± 0.023 | 0.724 ± 0.053 | 0.884 ± 0.017 | 0.858 ± 0.035 |
| **per 100 rounds** | **0.909 ± 0.021** | 0.880 ± 0.008 | **0.916 ± 0.038** | 0.931 ± 0.021 |
| per 1000 rounds | 0.903 ± 0.012 | **0.905 ± 0.019** | 0.866 ± 0.034 | **0.939 ± 0.005** |

Note: Experiments were performed repeatedly with three random seeds to report mean and standard deviation.
Note: * denotes statistically significant difference.

diagnosis of COVID-19. With the optimal configuration of ViT for modulation, it was possible to surpass the existing methods for distributed learning and achieve the performance comparable to data-centralized learning with our framework, even under the skewed data distribution. Moreover, our framework alongside clients to process multiple related tasks also improves the performances of individual tasks, while eliminating the need to share the data and large weights of the body network. These results suggest the suitability of the Transformer for collaborative learning in medical imaging and pave the way forward for future real-world applications.

## 6    Limitation and Potential Negative Societal Impacts

This work is more like a proof-of-the-concept study, rather than a ready-to-use solution for the industry. Therefore, it holds some limitations which may occur in a real-world application. Recent works on privacy attacks in the FL setting have implied that the belief that "Privacy can be protected by the decentralized nature of the FL" is not true [1, 56, 62]. In this work, however, the experimental results regarding the privacy issue such as threatening privacy via inversion attack are not suggested. Secondly, although the unique challenges of FL have been suggested by previous work including the problems of significant bottleneck in communication, stragglers, and fault tolerance which is exacerbated than in typical data-centralized learning [46, 27], we have not conducted the experiments to evaluate the robustness of our method against these unique challenges.

Although distributed learning-enabled learning without sharing data, decentralization is not a panacea against the privacy problem. Similar to the previously distributed learning methods, a potential risk arises from these limitations that our algorithm may not be free from privacy issues via model inversion attack against the server, since the server retains the parameters of the entire network in process of `FedAvg` of the heads and tails despite the split design of the sub-networks upon both client and server-sides. Although the risk could be mitigated to some degree with the specific settings (e.g. small gradient due to pre-trained backbone, deeper network, more pooling layer, a mixture of multiple tasks), the privacy problem should not be ignored since we aim to use this method in collaboration with hospitals where the patient privacy is a matter of the highest priority. Therefore, the methods to enhance the security such as differential privacy [33], secure multi-party computation [3], data compression [64] and authenticated encryption [39] as well as the recent method to prevent gradient inversion attack without sacrificing FL performance [48] should be utilized along with the proposed method to further reduce the risk of privacy leakage in a real-world application.

## Acknowledgements and Disclosure of Funding

This research was funded by the National Research Foundation (NRF) of Korea grant NRF-2020R1A2B5B03001980. This work was also supported by Institute of Information and communications Technology Planning and Evaluation (IITP) grant funded by the Korea government(MSIT) (No.2019-0-00075, Artificial Intelligence Graduate School Program(KAIST)), and by the KAIST Key Research Institute (Interdisciplinary Research Group) Project.

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
