# Supplementary Material:
# Federated Split Vision Transformer for COVID-19 CXR Diagnosis using Task-Agnostic Training

**Sangjoon Park**[1]*        **Gwanghyun Kim**[1]*

**Jeongsol Kim**[1]      **Boah Kim**[1]      **Jong Chul Ye**[1,2,3]

[1] Department of Bio and Brain Engineering
[2]Kim Jaechul Graduate School of AI, [3]Deptartment of Mathematical Sciences
Korea Advanced Institute of Science and Technology (KAIST)
{depecher, gwang.kim, wjdthf3927, boahkim, jong.ye}@kaist.ac.kr

## Abstract

This supplementary material discusses the details of implementation including hyperparameters, network configuration, and the framework protocol in Sec. A. It details the data-centralized and other distributed learning strategies used for comparison in Sec. B. The additional experimental results and ablation studies are provided in Sec. C and D. In Sec. E, details of hospital dataset are shown. Finally, ethic committee approval and permission informations are provided in Sec. F.

## A  Implementation Details

Here we describe the details of the hyperparameters used and their implementation. Our experiments were mainly implemented using Python version 3.7.5 and 3.8.8 with packages Pytorch version 1.7.1 and 1.8.1. We used a friendly federated learning framework (Flower) protocol [4] to implement the distributed learning system, Pytorch to implement the neural network.

### A.1  Hyperparameters

We have searched different hyperparameters for each chest X-ray (CXR) task. For the classification task, we used an SGD optimizer with a warm-up cosine learning rate scheduler with a max learning rate of 0.0005. For the segmentation task, we minimized the binary cross-entropy loss combined with dice and focal loss using Adam [11] with a learning rate of 0.0001 and cosine annealing scheduler with the maximum rounds of 2,000 for single-task and 1,000 for multi-task learning. Finally, the Adam optimizer along with the warm-up constant scheduler was used with the max learning rate of 0.00002. Gradient clipping was implemented, where the max gradient norm of 1.0 was chosen to enhance the rate of convergence, and the batch sizes per client were 2 for all tasks. We experimentally determined the optimal hyperparameters for each task. Table 1 and Table 2 provide the detailed hyperparameters used for each task during single-task learning and multi-task learning.

---

*Authors contributed equally.

Table 1: Hyperparameters used for single-task learning

| Hyperparameter | Classification | Segmentation | Detection |
|---|---|---|---|
| Learning rate (head and tail) | 0.0005 | 0.002 | 0.00002 |
| Learning rate (body) | 0.0005 | 0.0005 | 0.0005 |
| Scheduler | warm-up cosine | warm-up cosine annealing | warm-up constant |
| Number of rounds | 12,000 | 12,000 | 12,000 |
| Warm-up rounds | 500 | 500 | 500 |
| Mini-batch size | 2 per client | 2 per client | 2 per client |
| Maximum number of rounds | - | 2,000 | - |

Table 2: Hyperparameters used for multi-task learning

| Hyperparameter | Classification | Segmentation | Detection |
|---|---|---|---|
| Learning rate (head and tail) | 0.0005 | 0.002 | 0.00002 |
| Learning rate (body) | 0.0005 | 0.0005 | 0.0005 |
| Scheduler | warm-up cosine | warm-up consine annealing | warm-up constant |
| Number of rounds | 12,000 | 12,000 | 12,000 |
| Warm-up rounds | 500 | 500 | 500 |
| Mini-batch size | 2 per client | 2 per client | 2 per client |
| Maximum number of rounds | - | 2,000 | - |

## A.2 Details of network configuration

Fig. 1 depicts the details of network configurations of the proposed method for each task. For classification, the embedded feature of dimension $16 \times 16 \times 768$ from the head is first flattened into the dimension of $256 \times 768$, and used as the input after prepending a CLS token with the same hidden dimension to yield the input of dimension $257 \times 768$. The output from the Transformer body corresponding to this CLS token embed the comprehensive feature of the entire CXR image so that it can be used to make the final prediction (Fig. 1(a)). On the other hand, for the segmentation task, the features at the deepest level of TransUNet of the dimension $32 \times 32 \times 1024$ is used as the input of the Transformer after mapping into the dimension of $16 \times 16 \times 768$ and flattened to dimension of $256 \times 768$, and the CLS token is not utilized at all though it is prepended as the same way in the classification task to make the same feature size $257 \times 768$. The resulting transformed features from the body are mapped into original shape and utilized as the same in standard TransUNet architecture (Fig. 1(b)). Similarly, the model for the detection task doesn't use the CLS token, and it rather uses a similar approach to that of the segmentation task. The deepest level of the feature pyramid, which has features of the dimension $16 \times 16 \times 1024$, is first mapped into dimension of $16 \times 16 \times 768$, and is used as the input for the Transformer body after flattening and prepending CLS token to make the same dimension of $257 \times 768$ to other tasks. Then, the transformed feature from the body reverts to the original shape and position for the feature pyramid to be combined to yield the final output (Fig. 1(c)).

## A.3 Implementation of our framework upon Flower protocol

From the implementation perspective of this consequential process, the major hurdle for federated learning (FL) research is the paucity of open source frameworks that support scalable FL on multiple edge devices. Several studies performing FL on millions of edge devices have been published [9], but they are based on a closed industrial system developed by a private corporation and are not publicly available. Meanwhile, even though several open-source frameworks including Tensorflow federated [2], PySyft [1] and LEAF [6] enabled the experiments on FL simulation, they do not support heterogeneous clients, server-side orchestration and are neither scalable between multiple machines, nor language agnostic. Recently, an open-source framework, Flower has been developed to address this problem which supports the heterogeneous environment and scaling to multiple distributed clients. It offers stable, language- and deep learning framework-agnostic implementation. Moreover, it allows rapid adoption of the existing deep learning algorithm to evaluate their learning dynamics and performances in a federated setting. Therefore, we implemented our framework upon this Flower protocol.

Fig. 2 illustrates the core components of our framework based on Flower. Since the FL can be considered as an interplay between global (server) and local (client) computations, we implemented

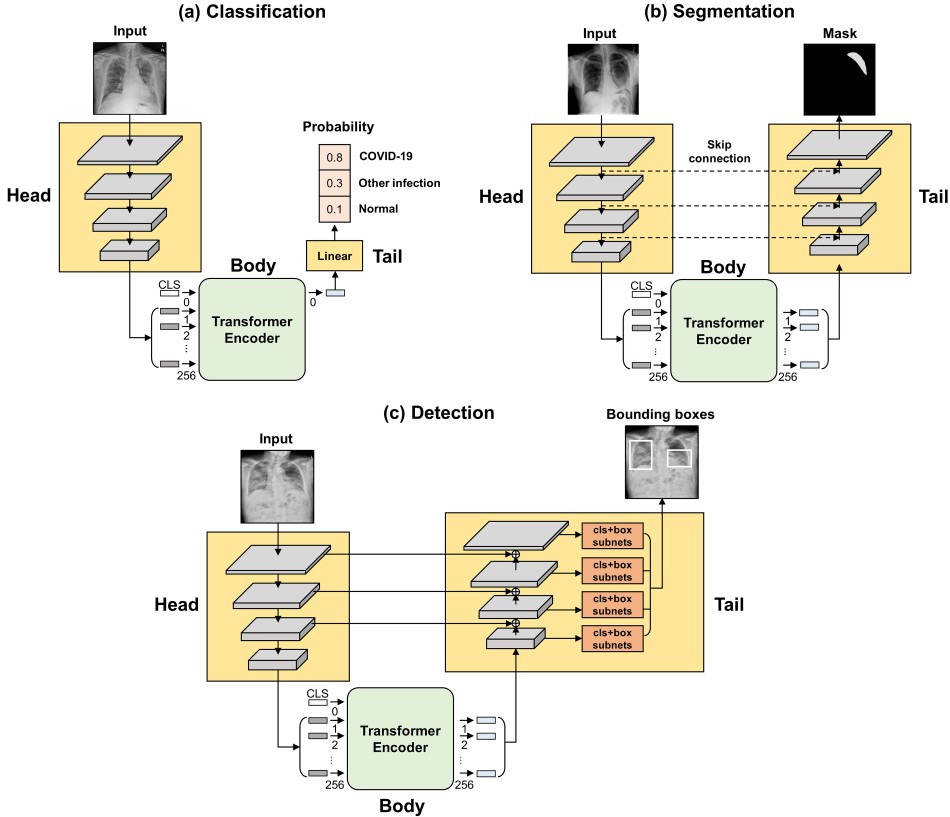

Figure 1: Detailed configuration for (a) classification, (b) segmentation and (c) detection tasks

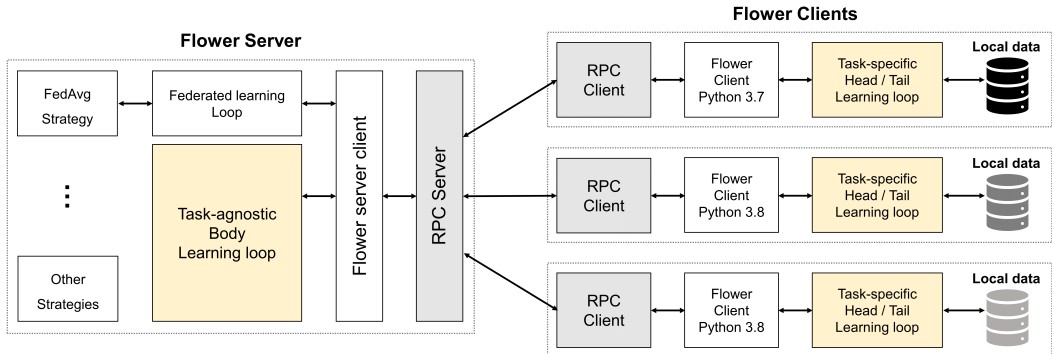

Figure 2: Implementation of our framework on top of Flower protocol.

the server and client-side components of our framework on top of the Flower server and clients. In Flower clients, task-specific loops of the heads and tails are performed with local data of each client, and the resulting features, gradients, and local parameters are passed toward the RPC client for communication. Then, the remote procedure call (RPC) client communicates with the RPC server in a language-agnostic manner using the bi-directional gRPC stream communication protocol [3], which offers an efficient binary serialization format. On the server-side, a task-agnostic body loop is performed using the features and gradients received. In addition, the aggregation, distribution of local parameters through a strategy such as federated averaging (`FedAvg`) are performed per averaging rounds. Finally, the features, gradients, and aggregated global parameters from the server revert to each client.

Different from previous studies that reported the result of single-device simulation [7], our method supports the simulation with multiple machines, which is close to real-world implementation of the system across the edge devices.

## B Data-centralized and Other Distributed Learning Methods

We perform the comparison of Federated Split Task-Agnostic (FESTA) with data-centralized and other distributed learning methods on the COVID-19 classification task which is the main task of this study. The details of each learning process are illustrated in Fig. 3.

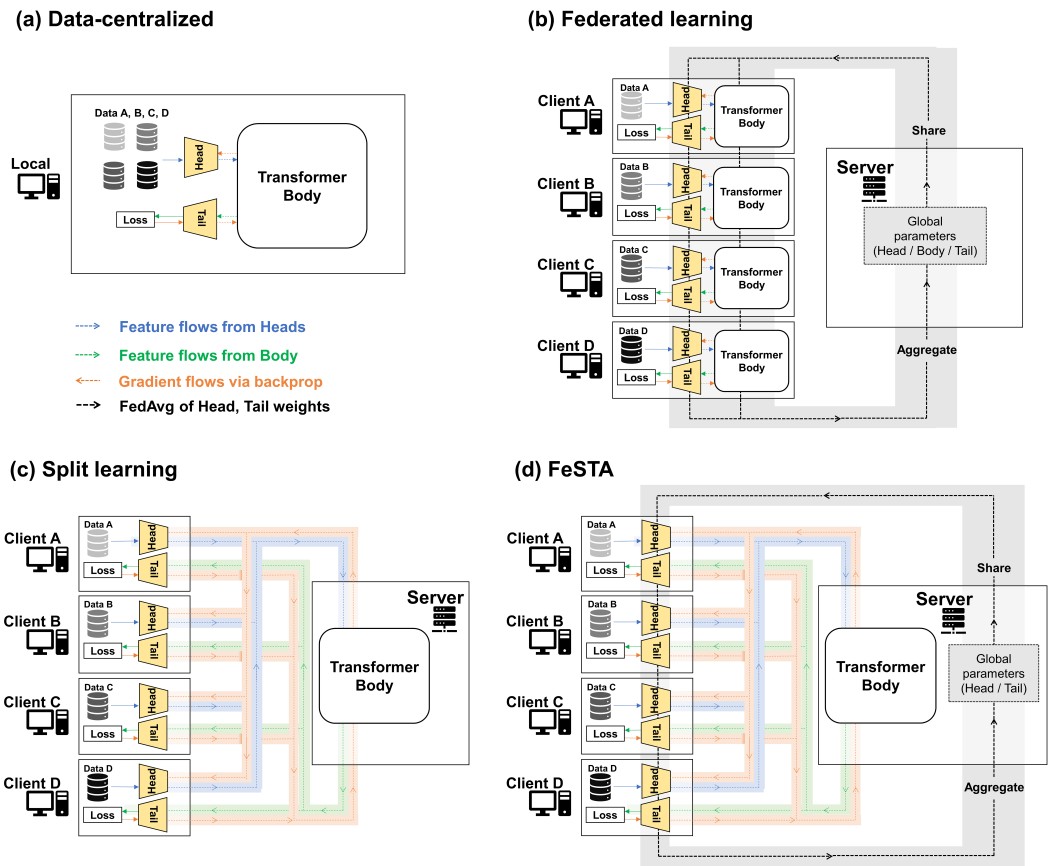

Figure 3: Detailed description for (a) data-centralized learning, (b) federated learning, (c) split learning, (d) FESTA learning strategies.

### B.1 Details of data-centralized learning

In data-centralized learning, the local data from six clients are centrally aggregated by the server and the single model is trained on a central server as represented in Fig. 3 (a). Batch size is set to 12 to match the setting of distributed learning strategies, accounting for batch size two per every six clients. Other settings are used as the same as FESTA for a fair comparison.

### B.2 Details of federated learning

In general, the simulation of the FL can be achieved by repeatedly doing three steps, as illustrated: i) update local parameters of the distributed model with local data on each client, ii) send the updated local parameters back to a server for aggregation, iii) distribute the aggregated model back to the clients for next rounds of local updates. Thus, we trained the entire network consisting of the head, body, and the tail is trained on each client with its local data in parallel without dividing it into

sub-networks components as in Fig. 3(b). This process can be formally written as in Algorithm 1. Regarding the experimental setting, the same settings to those of the proposed FESTA were used for comparison.

---

**Algorithm 1:** Federated learning

---

**1 Function** `ServerMain`:
2      Initialize the global weight $\bar{W}$ and distribute to each client
3      **for rounds** $i = 1, 2, \ldots R$ **do**
4          **for clients** $c \in \mathcal{C}$ **do in parallel**
5              $W_c \leftarrow$ `ClientUpdate`$(c)$
6          **if** $i \in$ UnifyingRounds **then**
7              Update $\bar{W} \leftarrow \frac{1}{N} \sum\limits_{c \in \mathcal{C}} W_c$
8              Distribute the global weight to client $W_c \leftarrow \bar{W}$ for each client $c \in \mathcal{C}$

**9 Function** `ClientUpdate`$(c)$:
10      $x_c, y_c \leftarrow$ Current batch of input & label from client $c$
11      $L_c \leftarrow \ell_c(y_c, \mathcal{T}_c(\mathcal{B}_c(\mathcal{H}_c(x_c))))$ & Backprop.
12      Update $W_c \leftarrow W_c - \eta \frac{\partial L_c}{\partial W_c}$
13      **return** $W_c$

---

**Algorithm 2:** Split learning

---

**1 Function** `ServerMain`:
2      Initialize the body weight $w_{\mathcal{B}}^{(1)}$ and client head/tail weights $(\bar{w}_{\mathcal{H}}, \bar{w}_{\mathcal{T}})$ in server
3      **for rounds** $i = 1, 2, \ldots R$ **do**
4          **for clients** $c \in \mathcal{C}$ **do in parallel**
5              **if** $i = 1$ **then**
6                  Set client $(w_{\mathcal{H}_c}^{(i)}, w_{\mathcal{T}_c}^{(i)}) \leftarrow (\bar{w}_{\mathcal{H}}, \bar{w}_{\mathcal{T}})$
7              $h_c^{(i)} \leftarrow$ `ClientHead`$(c)$
8              $b_c^{(i)} \leftarrow \mathcal{B}(h_c^{(i)})$
9              $\frac{\partial L_c^{(i)}}{\partial b_c^{(i)}} \leftarrow$ `ClientTail`$(c, b_c^{(i)})$ & Backprop.
10              `ClientUpdate`$(c, \frac{\partial L_c^{(i)}}{\partial h_c^{(i)}})$
11          Update body $w_{\mathcal{B}}^{(i+1)} \leftarrow w_{\mathcal{B}}^{(i)} - \frac{\eta}{N} \sum\limits_{c \in \mathcal{C}} \frac{\partial L_c^{(i)}}{\partial w_{\mathcal{B}}^{(i)}}$

**12 Function** `ClientHead`$(c)$:
13      $x_c \leftarrow$ Current batch of input from client $c$
14      **return** $\mathcal{H}_c(x_c)$
**15 Function** `ClientTail`$(c, b_c)$:
16      $y_c \leftarrow$ Current batch of label from client $c$
17      $L_c \leftarrow \ell_c(y_c, \mathcal{T}_c(b_c))$ & Backprop.
18      **return** $\frac{\partial L_c}{\partial b_c}$
**19 Function** `ClientUpdate`$(c, \frac{\partial L_c}{\partial h_c})$:
20      Backprop. & $(w_{\mathcal{H}_c}, w_{\mathcal{T}_c}) \leftarrow (w_{\mathcal{H}_c} - \eta \frac{\partial L_c}{\partial w_{\mathcal{H}_c}}, w_{\mathcal{T}_c} - \eta \frac{\partial L_c}{\partial w_{\mathcal{T}_c}})$

---

### B.3 Details of split learning

To simulate split learning (SL), we adopted the SL without label sharing as suggested in the original paper of SL [17]. The detailed process of the SL method used in our experiment can be presented as in Algorithm 2. The overall process of SL is similar to FESTA except for the fact that a step of aggregation and distribution by the central server is absent in SL as in Fig. 3(c). The splitting

configuration of head, body, and tail on client and server sides were the same as in the proposed FESTA. Since the local head and tail parameters of individual clients are not unified in SL, the inference results on the external testing dataset can be different between clients. Therefore, we calculated evaluation metrics for every six clients and averaged them to get the final score. The other experiment settings, including batch size and learning rate, remain the same as in the proposed FESTA.

## C  Additional Experiments

In this section, the results of additional experiments to further analyze the proposed FESTA learning method are suggested.

### C.1  Performances with increased number of COVID-19 cases

To provide more robust results using the larger corpus of data especially in terms of the number of COVID-19 cases, additional experiments were performed as follows.

We first swapped the hospital 1 data (containing 6 COVID-19 cases), which was originally used as the external test dataset, with the hospital 3 data (containing 80 COVID-19 cases), and repeated the experiments with the same setting. As suggested in Table 3, the proposed model retained stable performance in hospital 3 data with 80 COVID-19 cases.

Table 3: Number of COVID-19 cases with the different external set and the classification performances (AUC).

| External test set | COVID-19 cases | Average | COVID-19 | Others | Normal |
|:---:|:---:|:---:|:---:|:---:|:---:|
| Hospital 1 | 6 | $0.909 \pm 0.021$ | $0.880 \pm 0.008$ | $0.916 \pm 0.038$ | $0.931 \pm 0.021$ |
| Hospital 3 | 80 | $0.913 \pm 0.019$ | $0.871 \pm 0.043$ | $0.932 \pm 0.007$ | $0.935 \pm 0.015$ |

Note: Experiments were performed repeatedly with three random seeds to report mean and standard deviation.

Secondly, the additional analysis was performed by holding out all four hospital data (private) from the training set and by using them for external validation. Here, the label system had to be simplified into two categories, COVID-19 and non-COVID-19, as public datasets do not contain any label data for the "other infection" class. The proposed model presented stable performance even after excluding all private CXR data from the training set, dispelling the worry of data leakage problems. Although the performances were slightly decreased, it should be taken into consideration that the total amount of training data decreased to less than half of the original training data by removing the hospital 4 data (Table 4).

Table 4: Classification performances (AUC) of the proposed model using all four hospital datasets as an external testset.

| External test set | COVID-19 cases | COVID-19 |
|:---:|:---:|:---:|
| All four hospitals (hospital 1-4) | 94 | $0.879 \pm 0.043$ |

Note: Experiments were performed repeatedly with three random seeds to report mean and standard deviation.

Finally, we gathered additional anterior-posterior (AP) view CXR data labeled by the experts and combined them with the original posterior-anterior (PA) view data as shown in Table 5. The total amount of COVID-19 data has doubled, and COVID-19 cases in hospital 1 increased 6 to 81 CXRs. When adding the additional AP view CXRs and evaluating the performances in hospital 1 data, the performances of the proposed model were not compromised and rather increased especially for the diagnosis of COVID-19 as in Table 6.

### C.2  Comparison with task-specific expert and CNN-based multi-task learning models

Table 7 shows a comparison of the performances of each task between the proposed Transformer-based multi-task learning model trained with FESTA method and others. First, we compared the proposed MTL model with single task experts, defined as following for each task.

Table 5: Increased dataset and sources for COVID-19 diagnosis.

| CXR view | Total | Hospital 1 | Hospital 2 | Hospital 3 | Hospital 4 | NIH | Brixia | BIMCV |
|---|---|---|---|---|---|---|---|---|
| **AP view (added)** | | | | | | | | |
| Normal | 3662 | 97 | - | - | 117 | 3355 | - | 93 |
| Other infection | 204 | 19 | 76 | 92 | 17 | - | - | - |
| COVID-19 | 3322 | 75 | 278 | 213 | - | - | 2384 | 372 |
| Total AP CXRs | 7188 | 191 | 354 | 305 | 134 | 3355 | 2384 | 465 |
| | | | | | | | | |
| **All view (total)** | | | | | | | | |
| Normal | 17311 | 417 | 300 | 400 | 8978 | 7123 | - | 93 |
| Other infection | 1672 | 58 | 220 | 400 | 994 | - | - | - |
| COVID-19 | 5753 | 81 | 286 | 293 | - | - | 4313 | 780 |
| Total CXRs | 24736 | 556 | 806 | 1093 | 9972 | 7123 | 4313 | 873 |

Table 6: Number of COVID-19 cases after adding AP view CXRs and the classification performances (AUC).

| External test set | COVID-19 cases | Average | COVID-19 | Others | Normal |
|---|---|---|---|---|---|
| Hospital 1 (PA data) | 6 | 0.909 ± 0.021 | 0.880 ± 0.008 | 0.916 ± 0.038 | 0.931 ± 0.021 |
| Hospital 1 (PA and AP data) | 81 | 0.924 ± 0.006 | 0.943 ± 0.015 | 0.879 ± 0.007 | 0.949 ± 0.008 |

Note: Experiments were performed repeatedly with three random seeds to report mean and standard deviation.

- **Classification:** DenseNet-121 (D121) model with Probabilistic Class Activation Map operation Ye et al. [19]
- **Segmentation:** AlbuNet [14] based segmentation network (1st place model in Kaggle SIIM-ACR pneumothorax segmentation challenge [16])
- **Detection:** RetinaNet [12] model with SE-ResNext-50 encoder (2nd place model in Kaggle RSNA pneumonia detection challenge [13])

As provided in Table 7, the proposed MTL model outperformed the task-specific experts for each specific task. Of note, when the shared Transformer body was substituted with the shared convolutional neural network (CNN) layer for MTL, the performance was substantially dropped in the detection task. Combined together, the results demonstrated the value of the Transformer architecture leveraging global attention as well as local attention, which is suitable for MTL and cannot be substituted by other architecture like shared CNN layers.

Table 7: Comparison of performances with task-specific experts and CNN-based MTL models

| Tasks | Metrics | Task-specific experts | CNN-based MTL | Transformer-based MTL |
|---|---|---|---|---|
| **Classification** | AUC | 0.898 ± 0.004 | 0.907 ± 0.011 | **0.931 ± 0.004** |
| **Segmentation** | Dice | 0.736 ± 0.014 | 0.797 ± 0.018 | **0.821 ± 0.003** |
| **Detection** | mAP | 0.190 ± 0.006 | 0.159 ± 0.035 | **0.204 ± 0.002** |

Note: Experiments were performed repeatedly with three random seeds to report mean and standard deviation.

In addition, when compared with Kaggle's winning solutions available for the segmentation [16] and detection tasks [13], the proposed MTL model showed comparable performances as shown in Table 8, suggesting that the Transformer body do not deface the performances of the individual tasks.

Table 8: Comparison with Kaggle winning solutions for segmentation and detection tasks

| Segmentation | Dice | Detection | mAP |
|---|---|---|---|
| 1st place solution (description) | 0.764 ± 0.007 | 2nd place solution (SE-ResNext-50) | **0.211 ± 0.003** |
| 4th place solution (descrption) | **0.841 ± 0.004** | 2nd place solution (SE-ResNext-101) | 0.199 ± 0.003 |
| Proposed MTL model | 0.821 ± 0.003 | Proposed MTL model | 0.204 ± 0.002 |

Note: Experiments were performed repeatedly with three random seeds to report mean and standard deviation.

## C.3 Estimates of communication costs

With the intrinsic property of FESTA learning, a high computational burden is imposed on the server-side device, and this configuration is what we intended. Suppose, if most of the computation is performed on client-sides, all participating hospitals should have devices with high computational capacity. Forcing to prepare high computational resources for participants will obviously hinder the widespread adoption in a real-world application, and preparing a powerful server-side device with better security is rather practical. Nevertheless, there still remains a problem of computational costs between the server and clients. The communication costs between the server as the client can be estimated as follow.

When the period between averaging is $k$ and transmission of features, gradients, and network parameters are $F$, $G$ and $P$ respectively, total transmission from Server to Client $T$ can be represented as follows:

$$T = k \times (F + G) + P, \tag{1}$$

When parameter numbers of head, body, and tail are $P_h$, $P_b$ and $P_t$ respectively and $k$ is 100, $T$ for each distributed learning strategy can be formulated as follows:

$$T_{\text{FL}} = P_h + P_b + P_t, \tag{2}$$
$$T_{\text{SL}} = 100 \times (F + G), \tag{3}$$
$$T_{\text{FESTA}} = 100 \times (F + G) + (P_h + P_t), \tag{4}$$

If the transmission from Server to Client $T$ and that from Client to Server $T_{\text{C}\rightarrow\text{S}}$ are assumed to be equal ($T_{\text{C}\rightarrow\text{S}} = T$), total transmission $T'$ is as follows:

$$T' = 2T. \tag{5}$$

We then calculated the communication costs for feature/gradient transmission and parameter transmission per 1 averaging (=100 rounds) for each task as shown in Table 9. Despite the fact that the communication cost of the proposed FESTA framework was larger than that of SL, it was substantially lower than that of FL.

Table 9: Communication costs of the distributed learning methods during training per 1 averaging (=100 rounds)

| | Total transmission | Feature and gradient transmission | Network parameter transmission |
|---|---|---|---|
| **Classification** | | | |
| Federated learning | 159.365M | - | 159.365M |
| Split learning | 78.950M | 78.950M | - |
| FESTA | 105.580M | 78.950M | 26.630M |
| **Segmentation** | | | |
| Federated learning | 177.592M | - | 177.592M |
| Split learning | 78.950M | 78.950M | - |
| FESTA | 123.808M | 78.950M | 44.858M |
| **Detection** | | | |
| Federated learning | 226.450M | - | 226.450M |
| Split learning | 78.950M | 78.950M | - |
| FESTA | 172.665M | 78.950M | 93.715M |

## C.4 Statistical analysis

We also performed whether or not the performance gains with the Transformer architecture are statistically significant. As provided in Table 10, the proposed MTL model outperformed the model without the Transformer body with statistical significance, and the performance increase was more prominent in the MTL model.

Table 10: Statistical comparison of performances between the model with and without the transformer.

| | COVID-19 | | Others | | Normal | |
|---|---|---|---|---|---|---|
| | AUC (95% CI) | p-value | AUC (95% CI) | p-value | AUC (95% CI) | p-value |
| **w/o Transformer** | 0.867 (0.696 - 1.000) | - | 0.883 (0.817 - 0.948) | - | 0.889 (0.837 - 0.941) | - |
| **w Transformer (STL)** | 0.868 (0.749 - 0.987) | 0.988 | **0.905 (0.852 - 0.958)** | 0.498 | 0.927 (0.889 - 0.965) | **0.019** |
| **w Transformer (MTL)** | **0.945 (0.896 - 0.995)** | 0.266 | 0.893 (0.833 - 0.954) | 0.768 | **0.938 (0.903 - 0.974)** | **0.010** |

Note: For statistical comparison, p-values and Confidence Intervals (CIs) were calculated using DeLong's test.
Note: To evaluate the statistical significance, the models with medium performance were compared.

## D  Additional Ablation Studies in Multi-Task Learning Setting.

Additional ablation studies have been performed to examine the effect of Transformer body capacity and different training schemes, as shown in Table 11.

**Effect of Transformer body capacity**  We first evaluated the effect of the network capacity of the task-agnostic body model on the performances of individual tasks. Since the Transformer processes the task-agnostic modeling between features in a multi-task setting, there exists a possibility that the performance can further increase with the use of a dedicated server system with higher computational resources, once the model shows the performance proportional to the capacity of the Transformer body. As suggested in Table 11, the model equipped with a smaller body showed lower performance than that of a standard Transformer body equipped with 12 heads and 12 layers, suggesting the possibility of further improvement in performance with a Transformer with higher capacity.

**Training scheme**  Since our framework consists of the server-side and client-side sub-networks, it is possible to train only part of these sub-networks or train this sub-network after fixing the others. Thus, we experimented with various training schemes to evaluate their effect on the performance in multi-task settings. For the one-step learning approach, we trained the model after having all sub-networks, namely head, body, and tail, learnable for the entire training round. For the alternating approach, we alternately fixed and unfixed the parameters of the body and head/tail per 100 rounds. As shown in Table 11, both of these approaches show lower performance than the proposed two-step learning approach. This suggests that the simultaneous or alternating approach to train these sub-network components makes training unstable. Fixing the body for multi-task processing after certain rounds and fine-tuning the task-specific components may help to reach the better local minimum for the head and tail for each task, resulting in better generalization performance.

Table 11: Additional ablations in multi-task setting

| Tasks | Classification | Segmentation | Detection |
|---|---|---|---|
| | AUC | Dice | mAP |
| *Effect of Transformer body capacity* | | | |
| $H = 4, L = 4, D_{hidden} = 256$ | $0.916 \pm 0.011$ | $0.809 + 0.030$ | $0.200 \pm 0.007$ |
| $H = 8, L = 8, D_{hidden} = 512$ | $0.916 \pm 0.013$ | $0.826 + 0.001$ | $0.191 \pm 0.016$ |
| $\mathbf{H = 12, L = 12, D_{hidden} = 768}$ | $\mathbf{0.931 \pm 0.004}$ | $\mathbf{0.821 \pm 0.003}$ | $\mathbf{0.204 \pm 0.002}$ |
| *Effect of training strategy* | | | |
| One-step approach | $0.930 \pm 0.022$ | $0.801 \pm 0.024$ | $0.188 \pm 0.020$ |
| Alternating approach | $0.915 \pm 0.011$ | $0.799 \pm 0.021$ | $0.179 \pm 0.003$ |
| **Two-step approach** | $\mathbf{0.931 \pm 0.004}$ | $\mathbf{0.821 \pm 0.003}$ | $\mathbf{0.204 \pm 0.002}$ |

Note: Experiments were performed repeatedly with three random seeds to report mean and standard deviation.

## E  Details of Hospital Dataset

Table 12 describes the details about the CXR and clinical characteristics of four hospital data deliberately collected for this study.

Table 12: Details of CXR and patient characteristics of hospital datasets.

| Data | Hospital 1 | Hospital 2 |
|---|---|---|
| **CXR image details** | | |
| **Number of CXRs** | 365 | 452 |
| **Modality** | CR (93.7%), N/A (6.3%) | CR (99.8%), N/A (0.2%) |
| **Exposure time (msec)** | 6.7 ± 3.4 | 16.5 ± 7.7 |
| **Tube current (mA)** | 473.3 ± 198.1 | 307.8 ± 36.4 |
| **Bits** | 12 (12-14) | 12 (12-12) |
| **Clinical details** | | |
| **Age** | 45.8 ± 15.9 | 50.9 ± 17.7 |
| **Sex** | M (47.7%), F (45.2%), N/A (7.1%) | M (50.2%), F (48.8%) |
| **COVID-19 severity** | 1 (1-3) | 5.5 (1-6) |
| **CT positive cases** | N/A (100%) | N/A (100%) |
| **Country** | South Korea | South Korea |

| Data | Hospital 3 | Hospital 4 |
|---|---|---|
| **CXR image details** | | |
| **Number of CXRs** | 788 | 9838 |
| **Modality** | CR (100%) | CR (3.8%), DX (96.2%) |
| **Exposure time (msec)** | 11.6 ± 8.0 | 8.9 ± 3.9 |
| **Tube current (mA)** | 317.8 ± 30.7 | 298.9 ± 43.6 |
| **Bits** | 12 (12-14) | 14 (10-15) |
| **Clinical details** | | |
| **Age** | 46.9 ± 16.6 | 46.3 ± 14.4 |
| **Sex** | M (26.9%), F (34.0%), N/A (39.1%) | M (49.1%), F (47.0%), N/A (3.9%) |
| **COVID-19 severity** | 5 (1-6) | - |
| **CT positive cases** | Positive (3.8%), N/A (96.3%) | - |
| **Country** | South Korea | South Korea |

Abbreviations: CXR, chest X-ray; CR, computed radiography; DX, digital x-ray; M, male; F, female.
Note: Values are presented as mean ± standard deviation or median (range).

## F   Ethic Committee Approval and Permission Information.

**Ethic Committee Approval and Permission Information**   The four hospital data deliberately collected for COVID-19 classification were ethically approved by the Institutional Review Board of each hospital. According to their terms of use, the public datasets for the classification task (CheXpert [10], Valencian Region Medical Image Bank [BIMCV] [8], Brixia [15, 5], National Institutes of Health [NIH] [18]) can be used for research purposes. Likewise, the datasets for the segmentation tasks (SIIM-ACR pneumothorax segmentation challenge [16]) and the detection (RSNA pneumonia detection challenge [13]) can be used for academic research according to the terms of use.