# OpenReview forum: "Federated Split Task-Agnostic  Vision Transformer for COVID-19 CXR Diagnosis"
_NeurIPS.cc/2021/Conference — NeurIPS 2021 Poster_

### Official Review · Reviewer_N5nw · 2021-07-16

**Rating:** 6
**Confidence:** 4

**Summary:**

The authors proposed a multi-task learning framework with a shared feature encoder (body) and separate input embedding (head) and application-wise decoder (tail) components for each task in a distributed/federated learning setting. A number of multi-head self-attention modules from the transformer model are adopted here as the body part. The Federated Split Task-Agnostic Learning (FESTA) paradigm is proposed to compute/update the model parameters of head/tail parts on the client-side and the body parts on the server-side. A list of chest x-ray datasets is employed for the experiments, including datasets for COVID-19 classification (normal, pneumonia, and COVID-19), pneumothorax segmentation, and pneumonia detection. Superior results of the proposed method are reported in comparison to other learning paradigms. The manuscript is overall well-prepared, while several things could be improved and addressed(listed below).

**Limitations And Societal Impact:**

Yes

**Main Review:**

- One of the limits of the proposed method is that a large computation resource is required on the server-side.
- Additionally, the communication cost between client and server is more than doubled compared to the FedAvg algorithm, which also increases the risk of data privacy breaches.  The output of the Embedding model (the head, with both activations and gradients) could be used to reconstruct the inputs (patient images) easier than using only gradients in FedAvg.
- The implementation detail about the head, body, and tail parts for each task could be enhanced, e.g., what's the architecture of the tail part in the classification model, a single FC layer?
- Are the results show in Table 2 with a single task? Add multi-task ones as well?
- I am seriously concerned that the ablation studies are conducted using FedAvg, not the proposed method?
- On lines 171-174, I am not sure why the CheXpert dataset is introduced with much more details than other datasets while it is not even used in the experiments. And its label is not generated by experts but by an NLP-based labeler. On the other side, the authors should reveal more details about the data from 4 hospitals, e.g., geographic info.
- I will suggest the authors tune down the claim for the Non-iid settings since the datasets do not have the "extreme" non-iid setting as stated on line 246, and I did not see any particular design for the non-iid setting in this work.
- H, T for "det" and "seg" are mismatched with the input images in Fig. 1(lower left)
- Bolded results better suggest the best in the column.
- It will be helpful to discuss the relationship between the proposed method and the one listed below (first arxiv version on Dec. 2020). Both share a lot in the model architecture.
Chen H, Wang Y, Guo T, Xu C, Deng Y, Liu Z, Ma S, Xu C, Xu C, Gao W. Pre-trained image processing transformer. InProceedings of the IEEE/CVF Conference on Computer Vision and Pattern Recognition 2021 (pp. 12299-12310).

I will be happy to raise my ratings if the author could discuss/address some of the concerns.

**Time Spent Reviewing:**

5

---

> ### Author Response · Authors · 2021-08-09
> **Reply to Reviewer 4 (1/2)**
>
> > **One of the limits of the proposed method is that a large computation resource is required on the server-side.**
>
> In contrast to your concern, imposing high computational resources on the server-side rather than clients is exactly what we intended. The reason is that if most of the computation is performed on client-sides, all participating hospitals should have an environment with high computational resources. In the real-world application, forcing to prepare high computational resources to participate in hospitals is not realistic and therefore may impede the widespread application. Rather, preparing a powerful server with better security is more practical. Therefore, we think that the proposed method has considerable advantages in actual use.
>
> > **Additionally, the communication cost between client and server is more than doubled compared to the FedAvg algorithm, which also increases the risk of data privacy breaches. The output of the Embedding model (the head, with both activations and gradients) could be used to reconstruct the inputs (patient images) easier than using only gradients in FedAvg.**
>
> Thank you for the constructive comments. We calculated the communication costs for feature/gradient transmission and parameter transmission per 100 rounds for each task as in *Response Table 1*. Despite the fact that the communication cost of the proposed Federated Split Task-Agnostic (FESTA) framework was larger than that of split learning (SL), it was still lower than that of federated learning (FL). Actually, in our experiments, the longest time was required to train the same model with an FL method.
>
> **_Response Table 1. Communication costs of the distributed learning methods during training per 1 Federated Averaging (100 rounds)_**
>
> |  | Total | Feature and gradient | Network parameter |
> |:---|:---:|:---:|:---:|
> | **Classification** |  |  |  |
> | Federated learning | 159.365 M | - | 159.365 M |
> | Split learning | 78.950 M | 78.950 M | - |
> | FESTA | 105.580 M | 78.950 M | 26.630 M |
> | **Segmentation** |  |  |  |
> | Federated learning | 177.592 M | - | 177.592 M |
> | Split learning | 78.950 M | 78.950 M | - |
> | FESTA | 123.808 M | 78.950 M | 44.858 M |
> | **Detection** |  |  |  |
> | Federated learning | 226.450 M | - | 226.450 M |
> | Split learning | 78.950 M | 78.950 M | - |
> | FESTA | 172.665 M | 78.950 M | 93.715 M |
>
> - *Note: Detailed calculations of communication costs are provided in "Supplement: Calculations of communication costs" posted as an official comment.*
>
>
> As the reviewer pointed out, our method is not free from the privacy issue. That being said, this is a common problem across all methods using FL. Nonetheless, we believe that the actual risk of privacy leakage is not so large for the following reasons.
>
> * **Deeper network depth:** a previous study [1] concerning model inversion attack in FL simulates unrealistically shallow (e.g. 4-layer neural network) to facilitate successful inversion attack. However, in a deep network with more than a hundred layers, it will be more difficult to successfully reconstruct the image and the output will be more lossy. Since our model has more than a hundred layers (121 layers of Densenet head + 12 layers of transformer encoder body + 1 layer of tail), the lossless reconstruction of the input image will be very difficult by model inversion attacks.
> * **DenseNet backbone:** Recent studies about model inversion attack in FL uses ResNet model, but we used DenseNet backbone. As is well known, DenseNet contains more pooling layers than ResNet, which can be considered to be a more “lossy” architecture. In this kind of network, it is suspected that inversion may be less successful.
> * **Difficulty in attacking an already-trained network with high capacity:** As inversion attack leverages the gradient of losses, it is theoretically impossible to reconstruct the input from their gradient if the gradients of loss functions for different inputs are close to zero and therefore cannot be distinguished. Since we first pre-train the network backbone with a public dataset and then train the network using patient data with privacy, we observed that the network converges and the losses become small rapidly. In this setting, an inversion attack is suspected to be more difficult.
> * **Use of multi-task learning (MTL):** Our MTL setting uses a shared body layer, and different head and tail networks for individual tasks. In this setting, the global server receives the features from multiple clients handling different tasks, but the order of these features can be different between rounds due to network communication between server and clients. For instance, the server receives features in order of “detection”, “segmentation”, and “classification” in round 1, but it can receive the features in order of “segmentation”, “classification”, and “detection” in round 2. Since all features have the same shape, the attackers can be confused about which features are for which tasks. We expect that this setting will be less prone to an inversion attack.
>
> For these reasons, we expect that the privacy leakage problem of patients would be alleviated to some extent. Even under an inversion attack, due to the lossy processing by headers, it is expected that the reconstructed image will be of poor quality which is not good enough to threaten the patient's privacy.
> In addition, the aggregated head and tail parameters can be promptly removed from the global server after averaging to prevent inversion attacks in practical applications.
>
> > **The implementation detail about the head, body, and tail parts for each task could be enhanced, e.g., what's the architecture of the tail part in the classification model, a single FC layer?**
>
> Thanks for the constructive comments. For classification, we used a DesNet-121 based encoder (with Probabilistic Class Activation Map operation) as head and three binary classifiers (FC layer) for each class. For segmentation, head and tail are the convolutional neural network encoder and decoder of the standard U-Net. For detection, a SE-ResNext-50 encoder and feature pyramid network of RetinaNet was used as head and tail, respectively. Finally, the shared transformer body consists of 12 layers of standard transformer encoder as in vision transformer. We will add this detail in the revised version.
>
>
>
> --------------------------------------------------------------------------
>
> **References**
>
> [1] Geiping, Jonas, et al. "Inverting Gradients--How easy is it to break privacy in federated learning?." arXiv preprint arXiv:2003.14053 (2020).

---

> > ### Comment · Reviewer_N5nw · 2021-08-15
> > **Lean to accept**
> >
> > I appreciate the authors' initiative to reduce the computation load on the client-side, which should fit well for some scenarios but not all scenarios. However, In addition to the heavy server requirement, training on the server-side will also involve potential privacy leakage issues since network activations will also need to be transported in addition to gradients. It is highly feasible to recover the input images using only gradients with a small minibatch size (often < 4) as stated in the following paper:
> >
> > Yin H, Mallya A, Vahdat A, Alvarez JM, Kautz J, Molchanov P. See through Gradients: Image Batch Recovery via GradInversion. InProceedings of the IEEE/CVF Conference on Computer Vision and Pattern Recognition 2021 (pp. 16337-16346).
> >
> > I understand it is a common problem to all FL methods, but the proposed method has the potential to increase the risk(by transporting activations). I prefer the authors discuss it in the limitation.
> >
> > In all, many things are addressed in the authors' response. My final decision lean to accept if these could be added to the revised version.

---

> > > ### Author Response · Authors · 2021-08-17
> > > **RE: Lean to accept**
> > >
> > > We are glad to hear that our response addressed most of your concerns. Regarding the privacy issue, we agree with you that our method still has a privacy problem especially due to sharing of gradients between the clients and server [1]. Although the risk could be mitigated to some degree with the specific settings (e.g. small gradient due to pre-trained backbone), the privacy problem should not be ignored since we aim to use this method in collaboration with hospitals where the patient privacy is a matter of the highest priority. Therefore, the methods to enhance the security such as differential privacy [2], secure multi-party computation [3], data compression [4] and authenticated encryption [5] as well as the recent method to prevent gradient inversion attack without sacrificing FL performance [6] should be utilized along with the proposed method to further reduce the risk of privacy leakage in real-world application. We will discuss this context in the section 6, Limitation and Potential Negative Societal Impacts. Thank you for the comment.
> > >
> > >
> > >
> > > -----------------------------------
> > > **References**
> > > [1] Yin, Hongxu, et al. "See through Gradients: Image Batch Recovery via GradInversion." Proceedings of the IEEE/CVF Conference on Computer Vision and Pattern Recognition. 2021.
> > > [2] McMahan, H. Brendan, et al. "Learning differentially private recurrent language models." arXiv preprint arXiv:1710.06963 (2017).
> > > [3] Bonawitz, Keith, et al. "Practical secure aggregation for privacy-preserving machine learning." proceedings of the 2017 ACM SIGSAC Conference on Computer and Communications Security. 2017.
> > > [4] Zhu, Ligeng, and Song Han. "Deep leakage from gradients." Federated learning. Springer, Cham, 2020. 17-31.
> > > [5] Rogaway, Phillip. "Authenticated-encryption with associated-data." In Proceedings of the 9th ACM Conference on Computer and Communications Security, pp. 98-107. 2002.
> > > [6] Sun, Jingwei, et al. "Soteria: Provable Defense Against Privacy Leakage in Federated Learning From Representation Perspective." Proceedings of the IEEE/CVF Conference on Computer Vision and Pattern Recognition. 2021.

---

> > > ### Author Response · Authors · 2021-09-03
> > > **RE:RE: Lean to accept**
> > >
> > > Thank you for your time and efforts to review our paper. As you have already mentioned, we have addressed many concerns of reviewers in our responses. Could you please re-evaluate the rating based on our responses?

---

> ### Author Response · Authors · 2021-08-09
> **Reply to Reviewer 4 (2/2)**
>
>
> > **Are the results show in Table 2 with a single task? Add multi-task ones as well?**
>
> Thanks for the careful observation.  Per your suggestion, we added multi-task ones in *Response Table 2*.
>
> **_Response Table 2. Comparison for performance (AUC) of the proposed method with other strategies_**
>
>
> | Strategy | Average | COVID-19 | Others | Normal |
> |:---|:---:|:---:|:---:|:---:|
> | Data-centralized | 0.911 ± 0.016 | 0.883 ± 0.036 | 0.927 ± 0.013 | 0.923 ± 0.004 |
> | Federated learning | 0.891 ± 0.019 | 0.840 ± 0.035 | 0.926 ± 0.018 | 0.906 ± 0.028 |
> | Split learning | 0.863 ± 0.005 | 0.807 ± 0.012 | 0.892 ± 0.007 | 0.889 ± 0.019 |
> | FESTA (single-task learning) | 0.909 ± 0.021 | 0.880 ± 0.008 | 0.916 ± 0.038 | 0.931 ± 0.021 |
> | FESTA (multi-task learning) | 0.931 ± 0.004 | 0.926 ± 0.023 | 0.929 ± 0.016 | 0.938 ± 0.013 |
>
>
>
> > **I am seriously concerned that the ablation studies are conducted using FedAvg, not the proposed method?**
>
> We would like to assure the reviewer that the ablation studies were indeed conducted with FESTA. We used the notation FedAvg in *Table 5 (in the paper)*, since federated aggregation, averaging, and the distribution in FESTA is performed based on FedAvg. To avoid confusion of readers, we will revise notation in *Table 5 (in the paper)* from FedAvg to FESTA in the revised version.
>
> > **On lines 171-174, I am not sure why the CheXpert dataset is introduced with much more details than other datasets while it is not even used in the experiments. And its label is not generated by experts but by an NLP-based labeler. On the other side, the authors should reveal more details about the data from 4 hospitals, e.g., geographic info.**
>
> We agree with your comment. Accordingly, we have added more details about CXR and clinical characteristics of 4 hospital data deliberately collected for this study as shown in *Response Table 3*. It can be added in the revised version.
>
> **_Response table 3. Details of hospital datasets_**
>
>
> | Data | &nbsp; &nbsp; &nbsp; &nbsp; &nbsp; &nbsp; &nbsp;  Hospital 1 | &nbsp; &nbsp; &nbsp; &nbsp; Hospital 2 |   &nbsp; &nbsp;  &nbsp; &nbsp; &nbsp; &nbsp; &nbsp;  &nbsp;    Hospital 3 |   &nbsp;  &nbsp;   &nbsp;  &nbsp;  &nbsp;  &nbsp;  &nbsp;   Hospital 4 |
> |:---|:---:|:---:|:---:|:---:|
> | **CXR image details** |||||
> | Number of CXRs | 365 | 452 | 788 | 9838 |
> | Modality  | CR (93.7%), N/A (6.3%) | CR (99.8%), N/A (0.2%) | &nbsp; &nbsp; &nbsp; CR (100%) &nbsp; &nbsp; &nbsp; &nbsp; | CR (3.8%), DX (96.2%) |
> | Exposure time (msec) | 6.7 ± 3.4 | 16.5 ± 7.7 | 11.6 ± 8.0 | 8.9 ± 3.9 |
> | Tube current (mA) | 473.3 ± 198.1 | 307.8 ± 36.4 | 317.8 ± 30.7 | 298.9 ± 43.6 |
> | Bits | 12 (12-14) | 12 (12-12) | 12 (12-14) | 14 (10-15) |
> | **Clinical details** |  |  |  |  |
> | Age | 45.8 ± 15.9 | 50.9 ± 17.7 | 46.9 ± 16.6 | 46.3 ± 14.4 |
> | Sex | Male (47.7%), Female (45.2%), N/A (7.1%) | Male (50.2%), Female (48.8%) | Male (26.9%), Female (34.0%), N/A (39.1%) | Male (49.1%), Female (47.0%), N/A (3.9%)  |
> | COVID-19 severity | 1 (1-3) | 5.5 (1-6) | 5 (1-6) | - |
> | CT positive cases | N/A (100%) | N/A (100%) | Positive (3.8%), N/A (96.3%) | - |
> | Country | South Korea | South Korea | South Korea | South Korea |
>
> - *Note: Values are presented as mean ± standard deviation or median (range).*
> - *Note: For COVID-19 severity, the array-based annotation method suggested by Toussie et [2] was used.*
>
>
> > **I will suggest the authors tune down the claim for the Non-iid settings since the datasets do not have the "extreme" non-iid setting as stated on line 246, and I did not see any particular design for the non-iid setting in this work.**
>
> Thank you for your comment. We’ll tune down the claim.
>
> > **H, T for "det" and "seg" are mismatched with the input images in Fig. 1(lower left)**
>
> Sorry for the mistake. We’ll revise it.
>
> > **Bolded results better suggest the best in the column.**
>
> We will revise it according to your comments.
>
> > **It will be helpful to discuss the relationship between the proposed method and the one listed below (first arXiv version on Dec. 2020). Both share a lot in the model architecture. Chen H, Wang Y, Guo T, Xu C, Deng Y, Liu Z, Ma S, Xu C, Xu C, Gao W. Pre-trained image processing transformer. InProceedings of the IEEE/CVF Conference on Computer Vision and Pattern Recognition 2021 (pp. 12299-12310).**
>
> As in the paper mentioned by the reviewer, the easily decomposable design of the transformer has been gaining the attention of many researchers. Image processing transformer (IPT) [3] is one of the successful approaches in adopting the transformer to handle multi-task in computer vision, which has inspired our study quite a lot. However, in contrast to IPT which adopted a transformer encoder-decoder design to provide task-specific embeddings to the decoder, we used an encoder-only design similar to the original vision transformer model. In addition, based on our novel observation that this decomposable design is suitable not only for multi-task learning but also for distributed learning methods such as SL and FL, we devised the proposed method to maximally exploit the benefit from these key insights on the transformer, MTL and FL. We will emphasize this benefit in more detail in the final version.
>
> --------------------------------------------------------------------------
>
> **References**
> [2] Toussie, Danielle, et al. "Clinical and chest radiography features determine patient outcomes in young and middle-aged adults with COVID-19." Radiology 297.1 (2020): E197-E206.
>
> [3] Chen, Hanting, et al. "Pre-trained image processing transformer." Proceedings of the IEEE/CVF Conference on Computer Vision and Pattern Recognition. 2021.

---

> > ### Comment · Reviewer_N5nw · 2021-08-15
> > **Good to have the newly added results and data**
> >
> > Thank you for the newly added results and data, which I believe are important and helpful for the audience.

---

> > > ### Author Response · Authors · 2021-08-17
> > > **RE: Good to have the newly added results and data**
> > >
> > > We appreciate your helpful comments. They have significantly improved the details of this work.

---

> ### Author Response · Authors · 2021-08-09
> **Supplement: Calculations of communication costs**
>
> ### Total transmission per 1 Federated Averaging (FedAvg)
> To address reviewers' concern about the complexity, we provide an explicit formula that we can use to estimate the complexity of the FedAvg step used in the proposed method and comparative ones.  This calculation is used to obtain the results in *Response Table 1*.
>
> **Notation and Assumption**
> - Assume one image is used for the update of the network for 1 round.
> - $T'$ : Total transmission per 1 FedAvg
> - $T$ : Total transmission from Server to Client per 1 FedAvg
> - $k$ : Period between FedAvgs (= 100 rounds)
> - $F$ : Transmission size of feature per 1 round
> - $G$ : Transmission size of gradient per 1 round
> - $P$ : Transmission size of parameters per 1 FedAvg
> - $P_h$ : # of head parameters
> - $P_b$ : # of body parameters
> - $P_t$ : # of tail parameters
>
>
> **Total transmission from per 1 FedAvg**
> - Total transmission is the sum of transmission from Server to Client and transmission from Client to Server that is assumed to be equal.
> $$T'=2T$$
> - Total transmission from Server to Client is composed of feature and gradient transmission plus network parameter transmission as follows.
> $$T=  k\times(F + G)  + P$$
>
> - $T$ for each strategy can be fomulated as follows.
> $$ T_{FL} =  P_h + P_b + P_t $$
> $$ T_{SL} = 100 \times (F + G) $$
> $$ T_{FESTA} = 100\times(F+G) + (P_h + P_t) $$
>
>
>
> **In case of our network, $T$ for each strategy can be calculated as follows.**
> - For classification,
> $$ T_{FL}=  0.002M + 66.367M + 13.313M = 79.682M $$
> $$ T_{SL}= 100 \times (256\times768 + 256\times768)= 100 \times (0.197M + 0.197M) = 39.475M $$
> $$ T_{FESTA} = 100\times(0.197M + 0.197M) + (0.002M + 13.313M) = 52.790M $$
>
> - For segmentation,
> $$ T_{FL} =  7.387M + 66.367M + 15.041M = 88.796M  $$
> $$ T_{SL} = 100 \times (0.197M + 0.197M) = 39.475M $$
> $$ T_{FESTA} = 100\times(0.197M+0.197M) + (7.387M  + 15.041M ) = 61.904M $$
>
> - For detection,
> $$ T_{FL} =  27.085M + 66.367M + 19.773M = 113.225M   $$
> $$ T_{SL} = 100 \times (0.197M + 0.197M) = 39.475M $$
> $$ T_{FESTA} = 100\times(0.197M+0.197M) + (27.085M + 19.773M) = 86.333M $$
>
>
>
>
> **In conclusion, total transmission $T'$ can be calculated as follows.**
> - For classification,
> $$ T_{FL}' = 2T_{FL} =  159.365M $$
> $$ T_{SL}' = 2T_{SL} =  78.950M $$
> $$ T_{FESTA}' = 2T_{FESTA} =  105.580M $$
>
> - For segmentation,
> $$ T_{FL}' = 2T_{FL} =  177.592M $$
> $$ T_{SL}' = 2T_{SL} =  78.950M $$
> $$ T_{FESTA}' = 2T_{FESTA} =  123.808M $$
>
> - For detection,
> $$ T_{FL}' = 2T_{FL} = 226.450M $$
> $$ T_{SL}' = 2T_{SL} =  78.950M $$
> $$ T_{FESTA}' = 2T_{FESTA} = 172.665M $$

---

> > ### Comment · Reviewer_N5nw · 2021-08-15
> > **Heavy computation required on the server side?**
> >
> > Thank you for explaining the detailed data transmission between server and clients. I guess the transmission could be minimized as arranged by the authors. But I still concern about the efficiency and computational load on the server. For example, when the server is training the body part for one client, other clients' training requests are queued? The separate model training at least will increase the transmission times, which will also add the load on the server-side. I guess it will require better hardware for the multi-thread processing and heavy computation of training. On the contrary, regular FedAvg will not demand any training and only had a minimum hardware requirement of the server. I will suggest the author discuss this matter as the limitations.

---

> > > ### Author Response · Authors · 2021-08-17
> > > **RE: Heavy computation required on the server side?**
> > >
> > > Thank you for your constructive comments. In the proposed method, we parallelized the training of the server body as described in in Section 3.2 and Algorithm 1. In detail, the training of the server body is actually not queued for each client, but it is processed all at once after getting the intermediate features or gradients from all clients, by concatenating them like a batch. For this reason, the use of multiple clients (e.g. 10 clients) did not impose a problematic computational load on the server, even with the server of the general PC specification (CPU: Intel i7-9700K, DRAM: 64GB, GPU: 2 * NVIDIA RTX 3090).
> > > However, in this configuration, the straggler problem could be exacerbated, since the training of the server body can only be started after receiving the features or gradients from all clients. This is the reason why we adopted a setting that relies less on the computational resources of the clients, which can vary significantly between hospitals.
> > > Nonetheless, when the number of clients increases substantially to hundreds or more in real applications, the server-side overload problem may occur. Fortunately, Flower framework [1] that we used to implement our FESTA framework already provides the random sampling of clients when the number of participating clients is too large. By using this method, the training with more than hundreds of clients becomes feasible.
> > > Although we have tried to realistically reduce the computational load on the server-side by adopting above mentioned methods, we agree that there still exists a possibility of server overload due to the Transformer body on the server. Therefore, we will state the limitation in the final version.
> > >
> > >
> > >
> > > ---------------------------------------
> > >
> > > **References**
> > > [1] Beutel, Daniel J., et al. "Flower: A friendly federated learning research framework." arXiv preprint arXiv:2007.14390 (2020).

---

> ### Author Response · Authors · 2021-08-30
> **Thanks for suggesting that your final decision leans to accept**
>
> Dear Reviewer,
>
>
>
>
> Thanks again for your helpful review and suggestion that your final decision leans to accept.  With only a few days left for discussion, would you please let us know if you think further revision is needed to improve your ratings?
>
>
>
>
> Best regards, Authors

---

### Official Review · Reviewer_yJhi · 2021-07-16

**Rating:** 7
**Confidence:** 4

**Summary:**

This paper combines: split learning, federated learning, multi-task learning, vision transformers, and COVID-19 chest x-ray (CXR) classification. Specifically, the authors propose the Federated Split Task-Agnostic (FESTA) framework, motivated by privacy concerns of training models with patient level healthcare data (CXRs). This framework is for training large deep learning models based on private datasets distributed over clients (i.e. hospitals). Here, each client has a local copy of a "head" and "tail" of a vision transformer model, while the server has a shared body of a vision transformer. The training process is as follows:
1.) clients perform forward passes using their local data through their head networks and send the embedded representations to the server
2.) the server performs a forward pass with the embedded representations and sends the resulting representations back to their respective clients
3.) the clients finish the forward pass through their tail networks, compute the loss, and send gradients back through the server
4.) server continues backprop, client finishes backprop, everyone updates their weights (the server's network weights will be updated considering data from _all_ clients)
5.) occasionally, all the weights from the clients head and tail networks will be averaged together (FedAvg)

Additionally, the authors consider the case where some clients are training on different tasks (e.g. image classification, segmentation, detection) using the same modality (CXRs). Here, step 5 is applied over clients with the same task.

**Limitations And Societal Impact:**

The authors discuss this in Section 6 and suggest, "This work is more like a proof-of-the-concept study, rather than a ready-to-use solution for industry", while strongly motivating this work from an application standpoint (the COVID pandemic). This is flawed reasoning as "industry" will look to venues like NeurIPS for ideas, and the flaws from research published at NeurIPS (especially application focused papers) can be directly translated to society.

**Main Review:**

The main claims of this paper are:
1.) That the split-learning setup *and* federated-learing training setup result in models that are as good as training with a centralized dataset.
2.) That ViT models are "ideally suited" for the split learing/federated learning (FESTA) training setup.

Claim #1 is supported to some extent by the results shown in Table 2. In average performance the Data-centralized approach, FESTA, and federated learning perform within a stdev of each other, while Split learing performs worse.

Claim #2 is not supported. In Table 5 the authors show results with and without the "Transformer body" and both are within a standard deviation of each other. Further, the "without Transformer body" approach is not described. Does the server use a different model when a transformer body is not used? Would a CNN perform just as well?

The classification task setup uses 3 public datasets and 4 private datasets collected from hospitals / labeled by radiologists. One of the 4 private datasets is held out for evaluation. Data leakage in COVID-19 classification tasks has been a huge problem, with many preprints claiming wonderful model performance. The authors should consider holding out CXRs from all 4 hospitals as an experiment (especially if the same radiologists labeled all 4) to test the generalization capabilities of the models. This is a major limitation of the application of this work.

Finally, as the authors point out, split learning is not sufficient to guarantee data privacy, while the premise of the paper is a system that can be used in a federated manner among clients with private non-IID datasets. This is another major limitation of the application of this work.

Overall, this work presents an interesting application of a combination of different methodologies. Such systems will likely be necessary to train large medical image models that are able to generalize to new imagery. It needs to be improved with experiments demonstrating the value of transformers specifically, or rewritten (with experiments) to consider other architectures.   Because the paper is motivated by a privacy sensitive application, more discussion with potential solutions to the privacy problems in SL need to be added.

## Minor details

- The entire FESTA row is bolded in Table 2 even though the mean performance in the Others column is less than other values. Please specify what is bolded in Table captions.
- Be careful of copying text from [1] (cited as [33] in the text). I noticed that the data descriptions from these papers were very similar, e.g. sharing, "With a total of 224,316 CXR images from 65,240 subjects, the 32,387 lateral view images were excluded, leaving 29,420 PA and 161,427 AP view".

[1] Park, Sangjoon, Gwanghyun Kim, Yujin Oh, Joon Beom Seo, Sang Min Lee, Jin Hwan Kim, Sungjun Moon, Jae-Kwang Lim, and Jong Chul Ye. "Vision Transformer using Low-level Chest X-ray Feature Corpus for COVID-19 Diagnosis and Severity Quantification." arXiv preprint arXiv:2104.07235 (2021).

**Time Spent Reviewing:**

2

---

> ### Author Response · Authors · 2021-08-09
> **Reply to Reviewer 3 (1/2)**
>
> > **The main claims of this paper are: 1.) That the split-learning setup and federated-learing training setup result in models that are as good as training with a centralized dataset. 2.) That ViT models are "ideally suited" for the split learing/federated learning (FESTA) training setup.
> Claim #1 is supported to some extent by the results shown in Table 2. In average performance the Data-centralized approach, FESTA, and federated learning perform within a stdev of each other, while Split learing performs worse.
> Claim #2 is not supported. In Table 5 the authors show results with and without the "Transformer body" and both are within a standard deviation of each other. Further, the "without Transformer body" approach is not described. Does the server use a different model when a transformer body is not used? Would a CNN perform just as well?**
>
> We appreciate the reviewer’s comment. The “without Transformer body” approach means that the server conducts only the aggregation, averaging, and distribution of head and tail weights, without having its own shared body. This approach is identical to training the expert network (e.g. for classification, CNN model (DenseNet-121 model with Probabilistic Class Activation Map operation) specialized for chest X-ray (CXR) classification) with federated learning (FL). To address your comment, we performed additional analysis to evaluate the statistical difference between the models. Specifically, we performed the DeLong test to evaluate the statistical significance in the difference of AUCs between the models with and without the transformer body. As shown in *Response Table 1*, the model with the transformer body provided numerically better performance for all labels, and discriminating normal and abnormal CXRs was statistically significant (p<0.05). Overall, we can say that the transformer body enabled the performance increase in some labels, without deteriorating the performances in multiple tasks.
>
>
>
> **_Response Table 1. Statistical comparison of performances (AUC) between the model with and without the transformer_**
>
> |  | COVID-19 |  | Others |  | Normal |  |
> |:---|:---:|:---:|:---:|:---:|:---:|:---:|
> |  | **AUC (95% CI)** | **p-value** | **AUC (95% CI)** | **p-value** | **AUC (95% CI)** | **p-value** |
> | Without transformer | 0.867 (0.696 - 1.000) | - | 0.883 (0.817 - 0.948) | - | 0.889 (0.837 - 0.941) | - |
> | With transformer (single-task learning) | 0.868 (0.749 - 0.987) | 0.988 | 0.905 (0.852 - 0.958) | 0.498 | 0.927 (0.889 - 0.965) | 0.019 |
> | With transformer (multi-task learning) | 0.945 (0.896 - 0.995) | 0.266 | 0.893 (0.833 - 0.954) | 0.768 | 0.938 (0.903 - 0.974) | 0.010 |
>
>
>
>
> > **The classification task setup uses 3 public datasets and 4 private datasets collected from hospitals / labeled by radiologists. One of the 4 private datasets is held out for evaluation. Data leakage in COVID-19 classification tasks has been a huge problem, with many preprints claiming wonderful model performance. The authors should consider holding out CXRs from all 4 hospitals as an experiment (especially if the same radiologists labeled all 4) to test the generalization capabilities of the models. This is a major limitation of the application of this work.**
>
> As the reviewer suggested, we performed an additional analysis that held out CXR from all 4 hospital data (private) data from the training set and externally validated the generalization capability of the proposed model in these 4 hospital data. Here, the label system had to be simplified into two, namely COVID-19 and non-COVID-19, since public datasets do not contain the label for “other infection” as depicted in *Table 1 (in the paper)*. As provided in *Response Table 2*, the proposed model presented stable performance even after excluding all private CXR datasets from the training set, dispelling the worry of data leakage problems. Though performance itself slightly decreased, it should be taken into consideration that the total amount of training data decreased to less than half of the original data by removing the largest hospital 4 data.
>
>
>
> **_Response Table 2. Classification performances (AUC) of the proposed model using all 4 hospital datasets as an external testset._**
>
> | External test set | COVID-19 cases | COVID-19 |
> |:---|:---:|:---:|
> | Hospital  1-4 | 94 | 0.879 ± 0.043 |
>
>
>
> > **Finally, as the authors point out, split learning is not sufficient to guarantee data privacy, while the premise of the paper is a system that can be used in a federated manner among clients with private non-IID datasets. This is another major limitation of the application of this work.**
>
> As we already mentioned in the limitation, our method is not free from privacy issues. That being said, this is a common problem across all methods using FL. Nonetheless, we believe that the actual risk of privacy leakage is not so large for the following reasons.
>
> * **Deeper network depth:** a previous study [1] concerning model inversion attack in FL simulates unrealistically shallow (e.g. 4-layer neural network) to facilitate successful inversion attack. However, in a deep network with more than a hundred layers, it will be more difficult to successfully reconstruct the image and the output will be more lossy. Since our model has more than a hundred layers (121 layers of Densenet head + 12 layers of transformer encoder body + 1 layer of tail), the lossless reconstruction of the input image will be very difficult by model inversion attacks.
> * **DenseNet backbone:** Recent studies about model inversion attack in FL uses ResNet model, but we used DenseNet backbone. As is well known, DenseNet contains more pooling layers than ResNet, which can be considered to be a more “lossy” architecture. In this kind of network, it is suspected that inversion may be less successful.
> * **Difficulty in attacking an already-trained network with high capacity:** As inversion attack leverages the gradient of losses, it is theoretically impossible to reconstruct the input from their gradient if the gradients of loss functions for different inputs are close to zero and therefore cannot be distinguished. Since we first pre-train the network backbone with a public dataset and then train the network using patient data with privacy, we observed that the network converges and the losses become small rapidly. In this setting, an inversion attack is suspected to be more difficult.
> * **Use of multi-task learning (MTL):** Our MTL setting uses a shared body layer, and different head and tail networks for individual tasks. In this setting, the global server receives the features from multiple clients handling different tasks, but the order of these features can be different between rounds due to network communication between server and clients. For instance, the server receives features in order of “detection”, “segmentation”, and “classification” in round 1, but it can receive the features in order of “segmentation”, “classification”, and “detection” in round 2. Since all features have the same shape, the attackers can be confused about which features are for which tasks. We expect that this setting will be less prone to an inversion attack.
>
> For these reasons, we expect that the privacy leakage problem of patients would be alleviated to some extent. Even under an inversion attack, due to the lossy processing by headers, it is expected that the reconstructed image will be of poor quality which is not good enough to threaten the patient's privacy.
> In addition, the aggregated head and tail parameters can be promptly removed from the global server after averaging to prevent inversion attacks in practical applications.
>
>
> -----------------------------------------------------------
> **References**
>
> [1] Geiping, Jonas, et al. "Inverting Gradients--How easy is it to break privacy in federated learning?." arXiv preprint arXiv:2003.14053 (2020).

---

> ### Author Response · Authors · 2021-08-09
> **Reply to Reviewer 3 (2/2)**
>
> > **Overall, this work presents an interesting application of a combination of different methodologies. Such systems will likely be necessary to train large medical image models that are able to generalize to new imagery. It needs to be improved with experiments demonstrating the value of transformers specifically, or rewritten (with experiments) to consider other architectures. Because the paper is motivated by a privacy sensitive application, more discussion with potential solutions to the privacy problems in SL need to be added.**
>
> Despite the limitation, we expect that the actual risk of privacy leakage from attacks like model inversion would not be very high  in our method because of the aforementioned reasons.
>
> Regarding the experiments to demonstrate the value of the transformer, we conducted additional statistical analyses to find that the model equipped with the transformer body has better performance with statistical significance as shown in *Response Table 1*.
>
>
>
> **_Response Table 1. Statistical comparison of performances (AUC) between the model with and without the transformer_**
>
> |  | COVID-19 |  | Others |  | Normal |  |
> |:---|:---:|:---:|:---:|:---:|:---:|:---:|
> |  | **AUC (95% CI)** | **p-value** | **AUC (95% CI)** | **p-value** | **AUC (95% CI)** | **p-value** |
> | Without transformer | 0.867 (0.696 - 1.000) | - | 0.883 (0.817 - 0.948) | - | 0.889 (0.837 - 0.941) | - |
> | With transformer (single-task learning) | 0.868 (0.749 - 0.987) | 0.988 | 0.905 (0.852 - 0.958) | 0.498 | 0.927 (0.889 - 0.965) | 0.019 |
> | With transformer (multi-task learning) | 0.945 (0.896 - 0.995) | 0.266 | 0.893 (0.833 - 0.954) | 0.768 | 0.938 (0.903 - 0.974) | 0.010 |
>
>
>
>
> By adding the shared transformer body amidst the original model architecture, we enabled the MTL model to have better performances in individual tasks compared with single-task learning (STL) counterparts (*Table 3 in the paper*). When the shared transformer body was substituted with the shared CNN layer for MTL, the performance was significantly dropped in the detection task (*Response Table 3*) in our additional experiments. Combined together, these results demonstrate the value of transformer architecture leveraging global attention as well as local attention, which is suitable for MTL and cannot be substituted by other architecture like shared CNN layers.
>
>
> **_Response Table 3. Comparison of performances with task-specific expert networks and CNN-based multi-task learning (MTL) model_**
>
> | Tasks | Metrics | STL model | CNN-based MTL model | Transformer-based MTL model (ours) |
> |:---|:---:|:---:|:---:|:---:|
> | Classification | AUC | 0.909 ± 0.021 | 0.907 ± 0.011 | 0.931 ± 0.004 |
> | Segmentation | Dice | 0.798 ± 0.016 | 0.797 ± 0.018 | 0.821 ± 0.003 |
> | Detection | mAP | 0.202 ± 0.008 | 0.159 ± 0.035 | 0.204 ± 0.002 |
>
>
> >**The entire FESTA row is bolded in Table 2 even though the mean performance in the Others column is less than other values. Please specify what is bolded in Table captions.**
>
> > **Be careful of copying text from [1] (cited as [33] in the text). I noticed that the data descriptions from these papers were very similar, e.g. sharing, "With a total of 224,316 CXR images from 65,240 subjects, the 32,387 lateral view images were excluded, leaving 29,420 PA and 161,427 AP view".**
>
> We will revise them with caution. Thank you for the meticulous review.

---

> > ### Comment · Reviewer_yJhi · 2021-08-28
> > **response**
> >
> > Dear Authors,
> >
> > Thank you for such a detailed response.
> >
> > I am now convinced that the ViT model is providing some benefit for the task vs. the CNN. Further the 95% CIs on the per class/model AUC give a much better picture of the uncertainty of these models. As such, I am updating my score to a 7.

---

> > > ### Author Response · Authors · 2021-08-29
> > > **Thanks for increasing your rating!**
> > >
> > > Thank you for your positive feedback and for increasing your rating. We will ensure that all of the updated information contained in the rebuttal is incorporated into the final version of the paper.

---

### Official Review · Reviewer_1kwL · 2021-07-17

**Rating:** 7
**Confidence:** 4

**Summary:**

This paper presents a federated split task-agnostic (FESTA) framework that can take an effective use of the vision transformer's (ViT) to perform several CXR tasks simultaneously. The authors discover that the design of ViT is ideal for split learning, in which a network is designed into a client-server mechanism. They integrate ViT into their framework by combining the benefits of federated and split learning through its decomposable design. According to the authors, a model trained with the FESTA framework can leverage robust representations from multiple related tasks to improve individual task performance.

**Ethical Concerns:**

I would like the authors to make sure that the study has received ethics clearance and datasets are collected in accordance with guidelines and regulations from the third-party vendors (e.g., Kaggle).

**Limitations And Societal Impact:**

The authors acknowledge that this is more of a proof-of-concept study and that privacy concerns may arise. Furthermore, they have not done the necessary tests to evaluate the robustness of their approach against the problems of significant bottlenecks in communication, stragglers, and fault tolerance. However, because these issues are not the central theme of the work, they may be overlooked for the time being.

**Main Review:**

The work is somewhat a combination of a few existing techniques. The authors, however, make good use of previous studies to create a usable framework. They illustrate the usefulness of the framework by running comprehensive experiments on a variety of datasets and tasks.

The authors demonstrate that a model trained with the FESTA framework can leverage robust representations from multiple related tasks to improve individual task performance. The problem, however, the test set is quite a small one containing only 6 COVID images in the external/test set. Regarding this issue, I would like to see the result of the COVID classification task by considering ‘Hospital 3’ as an external/test set and ‘Hospital 1’ as part of the training/validation set.

The submission is clearly written and well-organized. However, it would strengthen the study and clarify the performance more precisely if the authors could compare their results with the winning solutions (e.g., segmentation and detection tasks) from the Kaggle platform. The readers could get a good understanding and performance of the proposed model compared to those of the Kaggle platform.

The findings are important, and I think the research community is likely to use or expand on them. Certainly, the submission does a better job of addressing a challenging task than previous work.

Overall, the study is an interesting one and the authors make an effort to support their claims with adequate experiments and promising results. However, the experimental setting for COVID classification, in my opinion, is not up to mark. According to the paper, the classification is the main task to achieve through FESTA, yet the test set comprises of only six COVID images. As a result, I am a bit skeptical about the robustness of the model with respect to the results. Nonetheless, considering the idea and the merit of the study, experiments conducted by the authors and performance shown in the manuscript, I believe this work is marginally above the threshold to accept. Hence, I vote for accepting the paper.

**Needs Ethics Review:**

Yes

**Time Spent Reviewing:**

8

---

> ### Author Response · Authors · 2021-08-09
> **Reply to Reviewer 2**
>
> > **The work is somewhat a combination of a few existing techniques. The authors, however, make good use of previous studies to create a usable framework. They illustrate the usefulness of the framework by running comprehensive experiments on a variety of datasets and tasks. The authors demonstrate that a model trained with the FESTA framework can leverage robust representations from multiple related tasks to improve individual task performance. The problem, however, the test set is quite a small one containing only 6 COVID images in the external/test set. Regarding this issue, I would like to see the result of the COVID classification task by considering ‘Hospital 3’ as an external/test set and ‘Hospital 1’ as part of the training/validation set.**
>
> Thank you for the constructive comment. Per your suggestion, we conducted an additional experiment by using hospital 3 as an external test set after integrating hospital 1’s data as part of the training/validation set instead. As shown in *Response Table 1*, the proposed model also showed stable performance in hospital 3 data with 80 COVID-19 cases.
>
>
>
> **_Response Table 1. The number of COVID-19 cases in the external set according to chest X-ray (CXR) views and the classification performances (AUC)._**
>
> | External test set | COVID-19 cases | Average | COVID-19 | Others | Normal |
> |:---|:---:|:---:|:---:|:---:|:---:|
> | Hospital 1 | 6 | 0.909 ± 0.021 | 0.880 ± 0.008 | 0.916 ± 0.038 | 0.931 ± 0.021 |
> | Hospital 3 | 80 | 0.913 ± 0.019 | 0.871 ± 0.043 | 0.932 ± 0.007 | 0.935 ± 0.015 |
>
>
>
> Furthermore, we gathered additional anterior-posterior (AP) view chest X-ray (CXR) data labeled by the experts to the original posterior-anterior (PA) dataset as shown in the *Response Table 2*. The total amount of COVID-19 data has doubled, and COVID-19 cases in hospital 1 increased from 6 to 81 images.
> Even after adding the additional AP view CXRs, the proposed model showed stable performance as shown in *Response Table 3*.
>
>
>
>
> **_Response Table 2. Increased dataset and sources for COVID-19 diagnosis_**
>
> | CXR view | Total | Hospital 1 (external) | Hospital 2 | Hospital 3 | Hospital 4 | NIH | Brixia | BIMCV |
> |:---|:---:|:---:|:---:|:---:|:---:|:---:|:---:|:---:|
> | **PA view (previously used)** |  |  |  |  |  |  |  |  |
> | Normal | 13649 | 320 | 300 | 400 | 8861 | 3768 | - | - |
> | Other infection | 1468 | 39 | 144 | 308 | 977 | - | - | - |
> | COVID-19 | 2431 | 6 | 8 | 80 | - | - | 1929 | 408 |
> | Total PA CXRs | 17548 | 365 | 452 | 788 | 9838 | 3768 | 1929 | 408 |
> | **AP view (newly added)** |  |  |  |  |  |  |  |  |
> | Normal | 3662 | 97 | - | - | 117 | 3355 | - | 93 |
> | Other infection | 204 | 19 | 76 | 92 | 17 | - | - | - |
> | COVID-19 | 3322 | 75 | 278 | 213 | - | - | 2384 | 372 |
> | Total AP CXRs | 7188 | 191 | 354 | 305 | 134 | 3355 | 2384 | 465 |
> | **All view (PA + AP view)** |  |  |  |  |  |  |  |  |
> | Normal | 17311 | 417 | 300 | 400 | 8978 | 7123 | - | 93 |
> | Other infection | 1672 | 58 | 220 | 400 | 994 | - | - | - |
> | COVID-19 | 5753 | 81 | 286 | 293 | - | - | 4313 | 780 |
> | Total CXRs | 24736 | 556 | 806 | 1093 | 9972 | 7123 | 4313 | 873 |
>
> - *Note: Hospital 1 data was used as an external test set, while other datasets were used for training and validation.*
>
>
>
>
>
> **_Response Table 3. The number of COVID-19 cases in the external set according to CXR views and the classification performances (AUC)._**
>
> | External test set | COVID-19 cases | Average | COVID-19 | Others | Normal |
> |:---|:---:|:---:|:---:|:---:|:---:|
> | Hospital 1 (PA data) | 6 | 0.909 ± 0.021 | 0.880 ± 0.008 | 0.916 ± 0.038 | 0.931 ± 0.021 |
> | Hospital 1 (PA and AP data) | 81 | 0.924 ± 0.006 | 0.943 ± 0.015 | 0.879 ± 0.007 | 0.949 ± 0.008 |
>
>
>
>
> > **The submission is clearly written and well-organized. However, it would strengthen the study and clarify the performance more precisely if the authors could compare their results with the winning solutions (e.g., segmentation and detection tasks) from the Kaggle platform. The readers could get a good understanding and performance of the proposed model compared to those of the Kaggle platform.**
>
> Per your suggestion, we have implemented and trained the winning solutions in the Kaggle platform using our dataset and compared their performance with the proposed method. As shown in *Response Table 4*, the proposed multi-task learning model showed comparable or better performances to the Kaggle winning solutions.
>
>
>
>
> **_Response Table 4. Comparison with Kaggle winning solutions for segmentation and detection tasks_**
>
> |  | Performance |  |
> |---|:---:|:---:|
> | **Segmentation** | **Dice** |  |
> | 1st place solution | 0.764 ± 0.007 |  |
> | 4th place solution | 0.841 ± 0.004 |  |
> | Ours | 0.821 ± 0.003 |  |
> | **Detection** | **mAP** |  |
> | 2nd place solution (SE-ResNext-50) | 0.211 ± 0.003 |  |
> | 2nd place solution (SE-ResNext-101) | 0.199 ± 0.003 |  |
> | Ours | 0.204 ± 0.002 |  |
>
>
>
> > **The findings are important, and I think the research community is likely to use or expand on them. Certainly, the submission does a better job of addressing a challenging task than previous work. Overall, the study is an interesting one and the authors make an effort to support their claims with adequate experiments and promising results. However, the experimental setting for COVID classification, in my opinion, is not up to mark. According to the paper, classification is the main task to achieve through FESTA, yet the test set comprises only six COVID images. As a result, I am a bit skeptical about the robustness of the model with respect to the results. Nonetheless, considering the idea and the merit of the study, experiments conducted by the authors, and performance are shown in the manuscript, I believe this work is marginally above the threshold to accept. Hence, I vote for accepting the paper.**
>
> According to the reviewer’s comment, we used Hospital 3 dataset which contains 80 COVID-19 cases as the external test set, and demonstrated the performance of our model in a larger number of COVID-19 cases  (see  *Response Table 1* above). Furthermore, we additionally collected AP view data for COVID-19 classification, which increase the total amount of data by two-fold. Especially for COVID-19 cases in the external test set, the number of COVID-19 cases increased from six to 81. In view-agnostic settings, both leveraging PA and AP images, our model showed even better performance with an increased number of cases as shown in *Response Table 3* above. This may result from the fact that the pneumothorax segmentation and the pneumonia detection tasks have already utilized AP view images, and unifying to utilize both PA and AP images for all three tasks improved learning of shared representation for the transformer architecture. This finding is impressive since the recent reviews on AI models for COVID-19 diagnosis claims that the AI models, which have been pouring out a lot recently, are not actually helpful at all from the viewpoint of real-world application [1], since the performances can also be affected by the factors like the views of CXR image [2].
>
> > **I would like the authors to make sure that the study has received ethics clearance and datasets are collected in accordance with guidelines and regulations from the third-party vendors (e.g., Kaggle).**
>
> As mentioned in Checklist sections 4 and 5, the deliberately collected datasets for the classification task are ethically approved by the Institutional Review Board. According to each term of use, the public datasets for the classification task (CheXpert [3], NIH [4], Brixia [5], BIMCV [6]) can be used for academic research purposes.  According to each competition rule (RSNA pneumonia detection challenge [7], SIIM-ACR pneumothorax segmentation challenge [8]), the datasets for the detection and segmentation can be used for academic research purposes.
>
> -----------------------------------------------------------------------------------------------------
>
> **References**
> [1] Wynants, Laure, et al. "Prediction models for diagnosis and prognosis of covid-19: systematic review and critical appraisal." bmj 369 (2020).
> [2] Roberts, Michael, et al. "Common pitfalls and recommendations for using machine learning to detect and prognosticate for COVID-19 using chest radiographs and CT scans." Nature Machine Intelligence 3.3 (2021): 199-217.
> [3] https://stanfordmlgroup.github.io/competitions/chexpert/
> [4] https://nihcc.app.box.com/v/ChestXray-NIHCC
> [5] https://brixia.github.io/
> [6] https://bimcv.cipf.es/bimcv-projects/bimcv-covid19/
> [7] https://www.kaggle.com/c/rsna-pneumonia-detection-challenge/rules
> [8] https://www.kaggle.com/c/siim-acr-pneumothorax-segmentation/rules

---

> > ### Comment · Reviewer_1kwL · 2021-08-17
> > **Robustness of the model and improved results.**
> >
> > I looked at other reviews as well as the authors' responses. The authors addressed my concerns for most of the cases. I would like the authors to update the results in the revised version if the paper finally gets accepted. Overall, I am inclined to accept the paper.

---

> > > ### Author Response · Authors · 2021-08-17
> > > **RE: Robustness of the model and improved results.**
> > >
> > > We sincerely appreciate your helpful review. Per your suggestion, we will update more robust and improved results in the revised version if it finally gets accepted.

---

> ### Author Response · Authors · 2021-08-30
> **Thanks for suggesting that your final decision leans to accept**
>
> Dear Reviewer,
>
>
>
> Thanks again for your helpful review and suggestion that your final decision leans to accept.  With only a few days left for discussion, would you please let us know if you think further revision is needed to improve your ratings?
>
>
>
> Best regards, Authors

---

> > ### Comment · Reviewer_1kwL · 2021-09-02
> > **Updated the original review and score**
> >
> > I have already mentioned that I am inclined to accept the paper; I have raised my previous score from 6 to 7 (Good paper, accept) in order to support the study.

---

> > > ### Author Response · Authors · 2021-09-02
> > > **Thanks for increasing your rating!**
> > >
> > > Thank you for your response and for increasing your rating.
> > > We will ensure that robust and improved results in this rebuttal will be incorporated int the final version of this paper if it finally get accepted.

---

### Official Review · Reviewer_Fz9S · 2021-07-20

**Rating:** 5
**Confidence:** 5

**Summary:**

This paper presents a federated learning (FL) framework for COVID-19 xray image analysis, which leverages the natural scalibility of visual transformer to establish a multi-task FL process. The proposed method jointly learns from different clients of various x-ray-based tasks including classification, detection and segmentation in a split learning manner, in which each client preserves their only transformer head and tail while jointly learning a shared transformer body maintained in the global server. Experimental results show that the established multi-task federated learning framework with transformer improves individual performances for each COVID-19 diagnosis task and are comparable with centralized training model.

**Limitations And Societal Impact:**

Authors have discussed the limitation and societal impact of this work.

**Main Review:**

1. Concern on technical novelty. This paper provides good insight of utilizing the natural configuration of transformer to promote multi-task federated learning by sharing the transformer body across tasks. However, the technical novelty of this paper are limited. Both the federated learning and split learning techniques in this paper are proposed previously, and the capacity of transformer for multi-task learning has also been well explored in many vision-language tasks. Though integrating transformer into federated scenario has some merits, it is insufficient to serve as the major contributions considering technical aspect.

2. Concern on privacy issue. The split learning procedure in this framework requires to send the data features of local clients into global server. Given that the networks parameters of local transformer head and tail are also attainable in global server for aggregation purpose, the sharing of samples features could lead to privacy leakage to some extent.

3. Transformer shows advantage on its flexibility which is the key enabling the establishment of multi-task federated learning, but does transformer also advantage on the task performance than the expert network on each task? I understand that authors utilize transformer as it enables to learn from different tasks under federated setting, but the architecture of transformer may not fit each task and could sacrifice the performance compared with federated training task-specific expert network at clients of the same task.

4. What is the criteria to select the global model for validation. Since the transformer body are shared among all clients, how do the authors determine the best model for testing? Will the performances at all three tasks be taken into consideration?

5. There are some mistakes in the experimental results, e.g., in Table 2, authors mark the proposed method with “0.916+-0.038” as bold in “Others”, but the baseline federated learning methods performs better “0.926+-0.018”.

6. It is unclear what the proposed method “W/o transformer body” mean. Without the server-side transformer body, does the remaining architecture only contain the transformer head and tail?

**Time Spent Reviewing:**

4 hours

---

> ### Author Response · Authors · 2021-08-09
> **Reply to Reviewer 1**
>
> > **Concern on technical novelty. This paper provides good insight of utilizing the natural configuration of transformer to promote multi-task federated learning by sharing the transformer body across tasks. However, the technical novelty of this paper is limited. Both the federated learning and split learning techniques in this paper are proposed previously, and the capacity of transformer for multi-task learning has also been well explored in many vision-language tasks. Though integrating transformer into federated scenario has some merits, it is insufficient to serve as the major contributions considering technical aspect.**
>
> As you pointed out, the major contribution of our paper comes from the key insight that the transformer is suitable for multi-task learning (MTL), by handling multiple tasks with its surprising flexibility not only in language tasks but also in vision tasks. In addition, we have recognized that this decomposable configuration of vision transformer architecture is also suitable for the combination of split learning (SL) and federated learning (FL). Though the concept of SL has already been proposed in 2018, there still remains a conundrum with convolutional neural network (CNN) based models that it is difficult to determine the proper layer of the neural network to split. Obviously, the decomposable head, body, and tail design of the transformer are well-suited for this scenario. With these insights, we proposed Federated Split Task-Agnostic (FESTA) framework, a learning method that utilizes both FL and SL for decomposable multi-task transformer models. We experimentally demonstrated that the benefit of FESTA framework can be maximally obtained with the transformer body, while the CNN-based body network provided poor performance for MTL as shown in *Response Table 1*.
>
>
>
> **_Response table 1. Comparison of performances with task-specific expert networks and CNN-based multi-task learning (MTL) model_**
>
> | Tasks | Metrics | Task-specific experts | CNN-based MTL model | Transformer-based MTL model (ours) |
> |:---|:---:|:---:|:---:|:---:|
> | Classification | AUC | 0.898 ± 0.004 | 0.907 ± 0.011 | 0.931 ± 0.004 |
> | Segmentation | Dice | 0.736 ± 0.014 | 0.797 ± 0.018 | 0.821 ± 0.003 |
> | Detection | mAP | 0.190 ± 0.006 | 0.159 ± 0.035 | 0.204 ± 0.002 |
>
>
>
> Despite the fact that our method is based on the existing techniques, we would like to claim our contribution in that our insight for the best combination of transformer, FL, and SL for MTL is important, by providing a methodology to utilize it to the maximum.
>
>
>
> > **Concern on the privacy issue. The SL procedure in this framework requires sending the data features of local clients into the global server. Given that the network parameters of the local transformer head and tail are also attainable in global servers for aggregation purposes, the sharing of samples features could lead to privacy leakage to some extent.**
>
> As we already mentioned in the limitation, our method is not free from privacy issues. That being said, this is a common problem across all methods using FL. Nonetheless, we believe that the actual risk of privacy leakage is not so large in the proposed method due to the following reasons.
>
> * **Deeper network depth:** a previous study [1] concerning model inversion attack in FL simulates unrealistically shallow (e.g. 4-layer neural network) to facilitate successful inversion attack. However, in a deep network with more than a hundred layers, it will be more difficult to successfully reconstruct the image and the output will be more lossy. Since our model has more than a hundred layers (121 layers of Densenet head + 12 layers of transformer encoder body + 1 layer of tail), the lossless reconstruction of the input image will be very difficult by model inversion attacks.
> * **DenseNet backbone:** Recent studies about model inversion attack in FL uses ResNet model, but we used DenseNet backbone. As is well known, DenseNet contains more pooling layers than ResNet, which can be considered to be a more “lossy” architecture. In this kind of network, it is suspected that inversion may be less successful.
> * **Difficulty in attacking an already-trained network with high capacity:** As inversion attack leverages the gradient of losses, it is theoretically impossible to reconstruct the input from their gradient if the gradients of loss functions for different inputs are close to zero and therefore cannot be distinguished. Since we first pre-train the network backbone with a public dataset and then train the network using patient data with privacy, we observed that the network converges and the losses become small rapidly. In this setting, an inversion attack is suspected to be more difficult.
> * **Use of MTL:** Our MTL setting uses a shared body layer, and different head and tail networks for individual tasks. In this setting, the global server receives the features from multiple clients handling different tasks, but the order of these features can be different between rounds due to network communication between server and clients. For instance, the server receives features in order of “detection”, “segmentation”, and “classification” in round 1, but it can receive the features in order of “segmentation”, “classification”, and “detection” in round 2. Since all features have the same shape, the attackers can be confused about which features are for which tasks. We expect that this setting will be less prone to an inversion attack.
>
> For these reasons, we expect that the privacy leakage problem of patients would be alleviated to some extent. Even under an inversion attack, due to the lossy processing by headers, it is expected that the reconstructed image will be of poor quality which is not good enough to threaten the patient's privacy.
> In addition, the aggregated head and tail parameters can be promptly removed from the global server after averaging to prevent inversion attacks in practical applications.
>
> > **Transformer shows an advantage on its flexibility which is the key to enabling the establishment of multi-task FL, but does transformer also advantage on the task performance than the expert network on each task? I understand that authors utilize transformer as it enables to learn from different tasks under federated setting, but the architecture of transformer may not fit each task and could sacrifice the performance compared with federated training task-specific expert network at clients of the same task.**
>
> To verify that the transformer architecture does not deteriorate the performance of individual tasks, we compared our multi-task model with task-specific experts trained via FL. The task-specific experts are defined as below:
> * **classification task:** DenseNet-121 (D121) model with Probabilistic Class Activation Map (PCAM) operation
> * **segmentation task:** AlbuNet based segmentation network (1st place model in Kaggle RSNA pneumothorax segmentation challenge [2])
> * **detection task:** RetinaNet model with SE-ResNext-50 encoder (2nd place model in Kaggle SIIM-ACR pneumonia detection challenge [3])
> As shown in the *Response Table 1* above, our MTL model equipped with the transformer outperformed task-specific expert models, showing that the transformer architecture does not compromise and rather provides astounding flexibility to benefit from MTL. Moreover, as shown in *Response Table 1* above, when the shared transformer body is substituted with the CNN-based body (consisting of 16 convolution modules with a larger amount of parameter [74 M] than the transformer body [66 M]), the performance of all tasks was substantially lower than the proposed model with the transformer body. Taken together, these results suggest that the transformer is indeed the key component enabling the establishment of multi-task FL.
>
> > **What is the criteria to select the global model for validation. Since the transformer body are shared among all clients, how do the authors determine the best model for testing? Will the performances at all three tasks be taken into consideration?**
>
> That’s one of the reasons why we divided the MTL process into two steps. As shown in *Figure 2 (in the paper)*, the body is trainable during the first step, but fixed during the second step. Since the body remains fixed throughout the second step, the best models (only head and tail) for each task can be selected even for different rounds (e.g. classification - model at 11000 round, segmentation - model at 11500 round, detection - model at 10500 round).
>
> > **There are some mistakes in the experimental results, e.g., in Table 2, authors mark the proposed method with “0.916+-0.038” as bold in “Others”, but the baseline federated learning methods performs better “0.926+-0.018”.**
>
> Thank you for your corrections. We will revise them in the revised version.
>
> > **It is unclear what the proposed method “W/o transformer body” means. Without the server-side transformer body, does the remaining architecture only contain the transformer head and tail?**
>
> Since we inserted the transformer body to the D121 model with PCAM operation as in *Figure 1(a) in the supplementary material* by dividing it into head and tail, the model architecture without a server-side transformer is simply the D121 model with PCAM operation. That is, the original D121 model with PCAM operation is trained via FL.
>
> We will clarify this in the revised version.
>
>
> ---------------------------------------------------------------------
> **References**
>
> [1] Geiping, Jonas, et al. "Inverting Gradients--How easy is it to break privacy in federated learning?." arXiv preprint arXiv:2003.14053 (2020).
> [2] https://www.kaggle.com/c/rsna-pneumonia-detection-challenge/
> [3] https://www.kaggle.com/c/siim-acr-pneumothorax-segmentation/

---

> ### Author Response · Authors · 2021-08-30
> **We are looking forward to your feedback**
>
> Dear Reviewer,
>
>
>
> Thanks again for your constructive comments. We would like to kindly remind you that we tried our best to respond to your concerns with additional experiments, etc. Given that there are approximately 3 days left for discussion, we would appreciate your timely feedback. Could you please go over our responses and let us know if there are any remaining issues?
>
>
>
> Best regards, Authors

---

### Review · Ethics_Reviewer_iUg6 · 2021-08-12

**Recommendation:**

The authors should engage more with the privacy criticism as they relate to the Vision Transformer either in section 6 or in section 2 on related work.

**Ethics Review:**

The ethics issues that arise are not a result of the instant research, but with systems of federated and split learning themselves. Though often touted as being better for privacy, these FL and SL systems are still vulnerable to attack. Further, decentralization is not a panacea against loss of privacy.

---

> ### Author Response · Authors · 2021-08-17
> **Reply to Ethics Reviewer**
>
> As the ethics reviewer pointed out, decentralization is not a panacea against the privacy problem. Per your suggestion, we’ll add the following context in the section 6:
>
> -	Although the distributed learning enabled learning without sharing data, decentralization is not a panacea against the privacy problem. Similar to the previous distributed learning methods, a potential risk arises from these limitations that our algorithm may not be free from privacy issues via model inversion attack against the server. Despite the fact that the settings used in our method (e.g. deeper network, more pooling layer, pre-trained network, mixture of multiple tasks) would make it difficult to reliably reconstruct the original image, there still remains a risk of privacy leakage. Therefore, privacy-preserving distributed learning methods such as differential privacy [1] or a recent method to prevent gradient inversion attack [2] should be considered with authenticated encryption [3] for real-world application.
>
>
> --------------------------------
>
> **References**
> [1] McMahan, H. Brendan, et al. "Learning differentially private recurrent language models." arXiv preprint arXiv:1710.06963 (2017).
> [2] Sun, Jingwei, et al. "Soteria: Provable Defense Against Privacy Leakage in Federated Learning From Representation Perspective." Proceedings of the IEEE/CVF Conference on Computer Vision and Pattern Recognition. 2021.
> [3] Rogaway, Phillip. "Authenticated-encryption with associated-data." In Proceedings of the 9th ACM Conference on Computer and Communications Security, pp. 98-107. 2002.

---

### Author Response · Authors · 2021-08-27
**Summary of the responses**

We thank all reviewers for the valuable suggestions with their effort and time. Your comments have greatly improved our work. Here we provide a summary of our responses, and please refer to the responses to each reviewer for more details.



**1. Compared our transformer-based multi-task learning (MTL) model with the single-task learning (STL) model, task-specific experts, and CNN-based MTL model.**
- Per the reviewers’ suggestions, we have extensively compared our transformer-based MTL model with other methods including the transformer-based STL model, task-specific expert networks, and CNN-based MTL model.
- As shown in our responses, the proposed model outperformed all the other methods, suggesting the suitability of the transformer within our MTL framework.



**2. More detailed discussion on the privacy issue.**
- As we have mentioned in the limitation, the proposed method is not free from the privacy issue similar to the other FL methods.
- Nonetheless, we believe that the actual risk of privacy leakage is not so large for our method due to the following reasons: Deeper network depth, DenseNet backbone, Pre-trained backbone weights, and the use of MTL. For these reasons, we expect that the privacy leakage problem of patients would be alleviated to some extent.
- However, we absolutely agree that more discussion should be added regarding this problem. We will add more details about this problem in the limitation section.



**3. More robust results with an increased number of COVID-19 cases.**
- Previously our results included only 6 COVID-19 cases in the external test dataset which is not sufficient to provide a robust result.
- Per a reviewer’s suggestion, we used the hospital 3 datasets (80 COVID-19 cases) as the external test dataset instead of the hospital 1 dataset (6 COVID-19 cases) and evaluated the performances, which also showed a similar diagnostic performance with the increased number of COVID-19 cases.
- In addition, we gathered additional CXRs cases with which the COVID-19 cases in hospital 1 increases from 6 to 81. With these additional cases, the proposed model showed stable and even better performance with an AUC of 0.943, which provides more robust results to support our claim.



**4. Comparison of communicational cost with other distributed learning methods.**
- We have compared the communicational cost of the proposed method with other distributed learning methods (e.g. federated learning and split learning).



**5. Details of data characteristics of four hospital datasets.**
- The details of the data characteristics (patient and CXR image characteristics) of four hospital datasets have been provided.

---

### Author Response · Authors · 2021-08-27
**The end of the discussion phase approaching**

Dear Reviewers,



Could you please go over our responses since we can discuss them with you only for the next week? We have faithfully responded to your comments and did our best to provide additional experimental results per your suggestions. We sincerely appreciate your time and efforts in reviewing our paper, and your constructive and insightful comments.



Thanks, Authors

---

### Decision · Program_Chairs · 2021-09-27

**Decision:**

Accept (Poster)

**Comment:**

The main concerns raised by the reviewers were: 1) comparison to multi-task, single-task, task-specific experts and CNN-based multi-task models. 2) more detailed discussions on the privacy issues 3) a discussion/comparison of communication costs. Overall, the reviewers found that the authors' response did a good job in addressing all these concerns. Indeed, the new results/discussion significantly strengthens the paper and should be included in a revised version.